

# Relative Timing of Uplift along the Zagros Mountain Front Flexure Constrained by Geomorphic Indices and Landscape Modelling, Kurdistan Region of Iraq

Mjahid Zebari[1.2], Christoph Grützner[1], Payman Navabpour[1], Kamil Ustaszewski[1]

[1]Institute of Geological Sciences, Friedrich-Schiller-University Jena, Jena, 07749, Germany.

[2]Geology Department, Salahaddin University-Erbil, Erbil, 44002, Kurdistan Region of Iraq.

*Correspondence to*: Mjahid Zebari (mjahid.zebari@uni-jena.de)

**Abstract.** The Mountain Front Flexure marks a dominant topographic step in the frontal part of the Zagros Fold-Thrust Belt. It is characterized by numerous active anticlines atop of an underlying basement fault. So far, little is known about the relative activity of the anticlines, about their evolution, and about how crustal deformation migrates over time. We assessed the relative landscape maturity of three along-strike anticlines (from SE to NW: Harir, Perat, and Akre) located on the hanging wall of the Mountain Front Flexure in the Kurdistan Region of Iraq to identify the most active structures and to get insights into the evolution of the fault and thrust belt. Landscape maturity was evaluated using geomorphic indices such as hypsometric curves, hypsometric integral, surface roughness, and surface index. Subsequently, numerical landscape evolution models were run to estimate the relative time difference between the onset of growth of the three anticlines, using the present-day topography of the Harir Anticline as a base model. A stream power equation was used to introduce fluvial erosion, and a hillslope diffusion equation was applied to account for colluvial sediment transport. For different time steps of model evolution, we calculated the geomorphic indices generated from the base model. While Akre Anticline shows deeply incised valleys and advanced erosion, Harir and Perat anticlines have relatively smoother surfaces and are supposedly younger than the Akre Anticline. The landscape maturity level decreases from NW to SE. A comparison of the geomorphic indices of the model output to those of the present-day Akre Anticline topography revealed that it would take the Harir Anticline 70±10 kyr and 200±20 kyr to reach the maturity level of the Perat and Akre anticlines, respectively, assuming constant erosion and rock uplift rates along the three anticlines. Since the factors controlling geomorphology (lithology, structural setting and climate) are similar for all three anticlines, and under the assumption of constant growth and erosion rates, we infer that uplift of the Akre Anticline started 200±20 kyr before that of the Harir Anticline, with the Perat Anticline showing an intermediate age. A NW-ward propagation of the Harir Anticline itself implies that the uplift has been independent within different segments rather than being continuous from NW to SE. Our method of estimating the relative age difference can be applied to many other anticlines in the Mountain Front Flexure region to construct a model of temporal evolution of this belt.



# 1 Introduction

The Zagros Fold-Thrust Belt is an active orogen that resulted from the collision between the Arabian and Eurasian plates and contains the deformed portions of the NE part of the former Arabian passive margin (Fig. 1; Alavi, 2007; Berberian, 1995; Mouthereau et al., 2012). Many aspects of the structural configuration and the evolution of the Zagros Fold-Thrust Belt are by now satisfactorily constrained, but the detailed spatial and temporal distribution of deformation across the belt is not yet well understood, especially in the NW part of the belt in Kurdistan Region of Iraq (KRI). The style, timing, and relative activity of front thrusts, deformation propagation, and along-strike variations have not been sufficiently studied, and neither is it well known which structures are currently the most active ones.

One of the morphologically most conspicuous structural elements of the Zagros Fold-Thrust Belt is the Mountain Front Flexure (MFF), which separates the High Folded Zone and the Foothill Zone (known in Iran as the Zagros Simply Folded Belt and Zagros Foredeep, respectively; Figs. 1 and 2; Alavi, 2007; Berberian, 1995; Jassim and Goff, 2006; McQuarrie, 2004; Mouthereau et al., 2012; Vergés et al., 2011). In most parts of the Zagros, the MFF marks a pronounced topographic step, separating folds with high amplitudes, narrow wavelengths, and higher topography in the High Folded Zone from folds with relatively low amplitudes, long wavelengths, and lower topography in the Foothill Zone (Fig. 2). The MFF is characterized by numerous active anticlines atop of fault strands emerging from a basement fault. It was suggested that the onset of the MFF activity in the NW Zagros was about 5±1 Ma based on low temperature thermochronology (Koshnaw et al., 2017). The timing of this activity is expected to differ along-strike the belt and, hence, the initiation of uplift of the anticlines on the hanging wall of the MFF is the key to understand this temporal and spatial evolution. In the neighbouring Iranian part, the MFF was a relatively long-lived structure active from 8.1 to 7.2 Ma to about the Pliocene-Pleistocene boundary. After that, only the southwesternmost anticline remained active in front of the MFF. This was inferred from progressive unconformities and magnetostratigraphy (Hessami et al., 2001, 2006; Homke et al., 2004).

In active orogens, the main factor that contributes to building up topography is ongoing convergence (Bishop, 2007; Burbank and Anderson, 2012; Whittaker, 2012). Recent advancements in the availability of high-resolution digital elevation models (DEMs) and GIS software allowed to quantitatively analyse the landscape (Bishop, 2007; Tarolli, 2014; Walcott and Summerfield, 2008). Tectonic geomorphology approaches and landscape maturity studies have been used extensively and proven to be efficient in studying the relative tectonic activity of different areas in contractional settings (Allen et al., 2013; Cheng et al., 2012; Mahmood and Gloaguen, 2012; Ramsey et al., 2008; Regard et al., 2009). Nevertheless, the NW part of the Zagros lacks modern studies on tectonic geomorphology with few exceptions. Bretis et al. (2011) detected sets of wind gaps (i.e. segments of river valleys abandoned due to lateral and vertical fold growth) in the High Folded Belt, NE of the MFF, suggesting that larger folds grew by linkage of smaller, shorter folds. Zebari and Burberry (2015) performed detailed analyses of various geomorphic indices for numerous anticlines in the High Folded Zone, concluding that the combination of clearly asymmetric drainage patterns and the mountain front sinuosity index (Bull, 2007; Keller et al., 1999) is a valuable tool for identifying putatively active fault-related folds. Obaid and Allen (2017) studied the landscape maturity of various anticlines



within the Zagros Foothill Zone and constrained the order of deformation of these anticlines by proposing an out-of-sequence propagation of underlying faults into the foreland. They proposed that the Zagros Deformation Front was among the earliest faults that have been reactivated within the Foothill Zone.

In an active orogen such as the Zagros, a better understanding of the temporal and spatial distribution of deformation due to ongoing tectonics can be achieved with landscape modelling. In the last two decades, numerical models have been extensively used to study landscape evolution (Chen et al., 2014; Tucker and Hancock, 2010; Valters, 2016; van der Beek, 2013) and several software packages were specifically developed for this purpose (Hancock et al., 2010; Hancock and Willgoose, 2002; Hobley et al., 2017; Refice et al., 2012; Salles and Hardiman, 2016; Tucker et al., 2001). Most of these models include algorithms for bedrock fluvial incision and hillslope creep as input parameters. Several studies have constrained the landscape evolution with the involvement of the corresponding tectonics and structures (Collignon et al., 2016; Cowie et al., 2006; Langston et al., 2015; Miller et al., 2007; Refice et al., 2012; Robl et al., 2008).

In this study, we assessed variations in the landscape maturity of three anticlines (from SE to NW, the Harir, Perat and Akre anticlines) located on the hanging wall of the MFF by quantitatively analysing landscape indices (hypsometric curve, hypsometric integral, surface roughness, and surface index) in order to distinguish more mature segments from less mature ones, and to reconstruct the relative variation of uplift time and/or rates along these anticlines. We then computed the difference in the onset of uplift between more mature anticlines and less mature ones using a landscape evolution model. The present-day topography of the least mature anticline served as an input model for computing the time that it takes this anticline to reach the same state as the most mature ones. Also, three structural cross-sections were constructed across the three anticlines to delineate their structural style and to link it with their landscape maturity.

## 2 Geological Setting

The Zagros Fold-Thrust Belt is the result of the collision between the Arabian and Eurasian plates (Fig. 1; Alavi, 2007; Berberian, 1995; Mouthereau et al., 2012). Continental collision started in the Early Miocene following the progressive subduction of Neo-Tethyan oceanic lithosphere underneath Eurasia (Agard et al., 2011; Csontos et al., 2012; Koshnaw et al., 2017; Mouthereau et al., 2012). The Zagros Fold-Thrust Belt extends for about 2000 km from the Strait of Hormuz in southern Iran to the KRI and further into SE Turkey. Since the onset of collision, the deformation front has propagated 250-350 km southwestward, involving the northeastern margin of the Mesopotamian foreland basin and the Persian Gulf into a largely NW-SE-trending foreland fold-thrust belt (Mouthereau, 2011; Mouthereau et al., 2007). The shortening across different sectors of the Zagros Fold-Thrust Belt is estimated to range between 10% and 32% (Blanc et al., 2003; McQuarrie, 2004; Molinaro et al., 2005; Mouthereau et al., 2007; Vergés et al., 2011). GPS-derived horizontal velocities between Arabia and Eurasia show present-day convergence rates between 19 and 23 mm/yr (McClusky et al., 2003). It is suggested that deformation partitioning occurs between the external and internal portions of the Iranian part of the Zagros Fold-Thrust Belt. While the internal Zagros Fold-Thrust Belt currently accommodates 3-4 mm/yr of right-lateral displacement along the Main Recent Fault (Fig. 1;





Reilinger et al., 2006; Vernant et al., 2004), the external part accommodates 7-10 mm/yr of shortening by thrusting and folding (Hessami et al., 2006; Vernant et al., 2004), 2-4 mm/yr of which is taken up by the MFF in the Fars Arc (Oveisi et al., 2009). However, no such estimates are available for the Iraqi segment of the Zagros Mountains. It is hence not known how much of the total Arabia-Eurasia plate convergence is being accommodated across the Iraqi part of the Zagros Fold-Thrust Belt.

5 The NW segment of the Zagros Fold-Thrust Belt in the KRI is subdivided to several NE-trending morphotectonic zones. These zones from NE to SW are: (i) Zagros Suture, (ii) Imbricated Zone, (iii) High Folded Zone and (iv) Foothill Zone (Figs 1 and 2; Jassim and Goff, 2006). These zones are bounded by major faults in the area. The faults include Main Zagros Thrust separating the Zagros Suture from the Imbricated Zone, High Zagros Fault that separate the Imbricate Zone from the High Folded Zone, and the Mountain Front Flexure that separate the High Folded Zone from the Foothill Zone (Figs 1 and 2; 10 Berberian, 1995; Jassim and Goff, 2006).

The deformed sedimentary succession is composed of 8 - 12 km thick Paleozoic to Cenozoic strata that rest on the Precambrian crystalline basement (Aqrawi et al., 2010; Jassim and Goff, 2006). The thick sedimentary cover consists of various competent and incompetent rock successions separated by detachment horizons. The infra-Cambrian Hormuz salt, which acts as a basal detachment in much of the Southern and Central Zagros Mountains in Iran, pinches out towards northwest (Hinsch and Bretis, 15 2015; Kent, 2010). Other intermediate detachment horizons influence the structural style of Central Zagros in Iran (e.g., Sherkati et al., 2005, 2006; Sepehr et al., 2006), but their behaviour is uncertain in NW Zagros due to limitations in outcrops and insufficient seismic profiles southwest of the Main Zagros Thrust. Some proposed detachment levels include Ordovician and Silurian shales (Aqrawi et al., 2010; De Vera et al., 2009), Triassic-Jurassic anhydrites (Aqrawi et al., 2010; Hinsch and Bretis, 2015; De Vera et al., 2009; Zebari, 2013; Zebari and Burberry, 2015), and Early Miocene anhydrite (Aqrawi et al., 20 2010; Csontos et al., 2012; Jassim and Goff, 2006; Kent, 2010; Zebari and Burberry, 2015).

The exposed geological units within the High Folded Zone are limited to c. 5 km thick Upper Triassic to Recent rocks (Fig. 2). Most anticlines are made up of Cretaceous carbonate rocks, while Upper Triassic-Lower Cretaceous strata are only exposed in the core of some anticlines. The Tertiary clastic rocks are preserved within the adjacent synclines. Within the studied structures, the Upper Jurassic-Lower Cretaceous Chia Gara and Lower Cretaceous Sarmord formations crop out in the core of 25 Bekhme and Zinta gorges only and consist of medium to thick bedded marly limestone, dolomitic limestone, and shale (Figs 2 and 3). The Lower Cretaceous succession of Qamchuqa and Upper Cretaceous Bekhme and Aqra formations consist of thick bedded and massive reef limestone, dolomitic limestone, and dolomite. These units are generally rigid and resistant to erosion. Thus, they build the raised cores of anticlines. The Upper Cretaceous-Tertiary succession consists primarily of clastic rocks, which are mostly denuded, and alternating Upper Paleocene and Upper Eocene limestone of Khurmala and Pila Spi formations, 30 respectively. They form a ridge surrounding the anticlines (Figs 2 and 3). Unconsolidated Quaternary sediments in the study area consist of slope deposits, residual soil, alluvial fan deposits, and river terraces.

There is no agreement concerning the overall structural style of the NW Zagros in KRI. Several authors (Al-Qayim et al., 2012; Ameen, 1991; Fouad, 2014; Jassim and Goff, 2006; Numan, 1997; De Vera et al., 2009) suggested that the Iraqi part of the Zagros Fold-Thrust Belt reveals a combination of both thin- and thick-skinned deformation. Partly relying on reflection



seismic data, it was also suggested that contraction has been localized on inherited passive-margin normal faults in the basement, which were inverted during the late stage of deformation since c. 5 Ma (Abdulnaby et al., 2014; Burberry, 2015; Koshnaw et al., 2017). The structural relief across the MFF (Fig. 2) is likely linked to blind thrusts in the basement (Al-Qayim et al., 2012; Ameen, 1991, 1992; Fouad, 2014; Koshnaw et al., 2017; Numan, 1997; De Vera et al., 2009). The same linkage

between structural relief and a regional basement blind thrust is also documented in the Iranian Zagros (Blanc et al., 2003; Emami et al., 2010; Leturmy et al., 2010; Sherkati et al., 2006). Alternatively, Hinsch and Bretis (2015) argued that the structural relief in the hanging wall of the MFF is related to an underlying duplex structure that is linked to a stepped detachment horizon rooting in an early Paleozoic detachment in the internal parts of the orogen. The relief has been attributed to the accumulation of the Hormuz salt in the Iranian Zagros (McQuarrie, 2004). Even though the MFF is believed to be a

major blind thrust in the basement (Berberian, 1995), it is usually mapped along the southwestern limb of the last high anticline where the Pila Spi limestones or the Bekhme and Aqra limestones crop out (Fouad, 2014; Jassim and Goff, 2006; Numan, 1997). Given that landforms in the vicinity of the MFF indicate ongoing tectonic deformation, we suspect that these blind faults might be active at present. Unfortunately, however, instrumental seismicity in the entire region is too diffusely distributed to be attributed to any particular faults (Jassim and Goff, 2006).

Structurally, this segment of Zagros Fold-Thrust Belt is dominated by NE-SW trending fault-related folds, the trend of folds changes to nearly E-W to the west of the Greater Zab River (Fig 2). The folds are usually S-verging and the related faults emerge to the surface within both Imbricated Zone and High Folded Zone, while they remain blind within the Foothill Zone (Fouad, 2014; Hinsch and Bretis, 2015). This is also seen within the studied anticlines, which have thrust fault in their forelimb. The Perat Anticline has a thrust in its back limb as well (Fig 4).

**3 Data and Methods**

**3.1 Geomorphic Indices**

The present-day relief in the study area resulted from a competition between rock uplift triggered by horizontal contraction and erosion destroying it. Parameters controlling these competing processes are the rate of tectonic accretion, rock erodibility and climate (Bishop, 2007; Burbank and Anderson, 2012).

In order to quantitatively analyse the landscape for the Harir, Perat and Akre anticlines (Figs 2 and 4), we determined four geomorphic indices: (i) hypsometric curve, (ii) hypsometric integral, (iii) surface roughness, and (iv) surface index. These are considered proxies for the relative maturity of a particular landscape. The hypsometric curve and the hypsometric integral highlight raised and flat surfaces. The surface roughness value is mainly sensitive to incision (Andreani et al., 2014; Andreani and Gloaguen, 2016; Pike and Wilson, 1971); the surface index is a measure for the amount of erosion. When referring to the

results obtained by using this set of geomorphic indices, we colloquially refer to them as "landscape maturity" parameters.



### 3.1.1 Hypsometric curve

The hypsometric curve for a basin is the frequency distribution of elevation of the watershed area below a given height (Strahler, 1952). Convex-shaped hypsometric curves represent less mature stages of the basin while concave-shaped curves represent older stages (Ohmori, 1993; Pérez-Peña et al., 2009). Hypsometric curves are usually calculated for a specific

drainage basin. However, in this study we calculated them for entire anticlinal ridges, restricting our considerations to those areas where Upper Cretaceous carbonates are exposed (Fig. 5). This allowed us to make realistic comparisons between the three anticlines, neglecting the differences in rock erodibility that arise when varying lithologies are included. Wind gaps and water gaps as well as the plunging crests of the anticlines were also excluded from the calculation.

### 3.1.2 Hypsometric integral

The hypsometric integral (HI) illustrates the distribution of landmass and it marks the isolated upraised mass above a relatively plain surfaces of low values from poorly eroded, broad and plain surface of high values. The hypsometric integral is computed for a certain area by the following equation (Pike and Wilson, 1971):

$$HI = \frac{h_{mean} - h_{min}}{h_{max} - h_{min}}$$
(1)

where $h_{mean}$, $h_{min}$ and $h_{max}$ are the mean, minimum and maximum elevations [m] of the examined area.

### 3.1.3 Surface roughness

The surface roughness (SR) measures how much an area deviates from being totally flat. It differentiates flat planar surfaces with values close to 1 from irregular surfaces with higher values. It increases with the increase in incision by streams. The surface roughness is calculated using the following equation (Grohmann, 2004):

$$SR = \frac{TS}{FS}$$
(2)

where TS and FS are the areas [m²] of the actual topographic surface and the corresponding projection of that surface onto a planar surface, respectively.

### 3.1.4 Surface index

The surface index (SI; Andreani et al., 2014) combines elevations, hypsometric integral and surface roughness to map simultaneously preserved and eroded portions of an elevated landscape. It is calculated using equation 3 (Andreani and

Gloaguen, 2016):

$$SI = (N_{HI} * N_h) - N_{SR}$$
(3)




where $N_{HI}$, $N_h$ and $N_{SR}$ are the normalized elevations, hypsometric integral and surface roughness values, respectively. Elevated and poorly incised landscapes with high hypsometric integral and low surface roughness show positive surface index values. Highly dissected landscapes with a high surface roughness yield negative surface index values. This means that the surface index is also sensitive to elevation.

**3.1.5 Digital elevation models**

The geomorphic indices for this study were calculated from the 12 m resolution TanDEM-X DEM (Krieger et. al., 2007) obtained from the German Aerospace Center (DLR) and the 30 m resolution SRTM1 DEM (NASA JPL, 2013); these two inputs were used since different DEM inputs give slightly different geomorphic results (Andreani et al., 2014; Koukouvelas et al., 2018; Obaid and Allen, 2017). Geomorphic indices were calculated using both the TanDEM-X and the SRTM1 data.

However, the TanDEM-X data revealed numerous artefacts and voids, which made calculations unstable and results unreliable. All results of the geomorphic indices and all subsequent calculations presented in the following sections were calculated from a 100 x 100 pixel (3 x 3 km) moving window on the 30 m resolution SRTM1 data. A larger moving window makes the obtained measurements smoother and vice versa. The size of the moving window must be chosen based on the scale of the target; here we targeted anticlines with wavelengths varying from 5 to 8 km. A 3 km moving window covered

almost an entire limb of an anticline. The calculations within the moving window were performed using the neighbourhood toolset in ESRI ArcGIS 10.4 software.

**3.2 Landscape Evolution Model**

We built a landscape evolution model to quantify the time difference in between the maturity level of the Akre and Harir anticlines by comparing the geomorphic indices of the evolved landscape with those of both anticlines based upon the open-

source Landlab toolkit (Hobley et al., 2017; http://landlab.github.io).

We used two components in our model: one simulating erosion due to fluvial action and another simulating sediment transport along slopes due to soil creep. Chen et al. (2014) showed that consideration of only these two components is sufficient for many landscapes but cannot model sedimentation. However, from field observations we know that no significant sedimentation takes place on the slopes of the analysed anticlines. In slopes on anticline flanks, the detachment-limited erosion due to the

fluvial system tends to be the dominant process (Howard, 1994). To detect changes in the landscape due to fluvial erosion through time, we applied the commonly accepted idea that the rate of stream incision is directly proportional to the hydraulic shear stress of a stream (Braun and Willett, 2013). Consequently, we used the stream power incision law (Sklar and Dietrich, 1998; Whipple and Tucker, 1999):

$$\frac{\partial z}{\partial t} = KA^m S^n \tag{4}$$



where ∂z/∂t is the erosion rate [myr$^{-1}$]; K is an erodibility coefficient [yr$^{-1}$m$^{(1-2m)}$] that encompasses the influence of climate, lithology, and sediment transport processes; A is the upstream drainage area [m$^2$] and is typically taken as a proxy for discharge (Wobus et al., 2006); S = ∂z/∂x is the local channel slope [m/m]; z is the elevation [m]; and m and n are the area and slope exponents, respectively. The stream power incision law (Eq. 4) is derived since the upstream drainage area A scales with channel discharge and channel width. The magnitude of the sediment flux in the channel is assumed to equal unity in the standard detachment-limited stream power model (Perron, 2017; Whipple, 2002). In the model, an incision threshold (C) was included, below which no incision occurs (Hobley et al., 2017).

To account for the provision of sediment due to soil creep from slopes outside the river system, we used the hillslope diffusion equation (Culling, 1963; Tucker and Bras, 1998):

$$\frac{\partial z}{\partial t} = K_d \nabla^2 z \tag{5}$$

where $K_d$ is the diffusivity coefficient [m$^2$yr$^{-1}$], z is the elevation [m], and $\nabla^2$ is the Laplace operator, i.e. the divergence of the gradient.

Finally, the overall evolution of the landscape in different time steps was calculated as the uplift rate minus the changes due to both fluvial erosion and the hillslope diffusion (Temme et al., 2017):

$$\frac{\partial z}{\partial t} = U - KA^m S^n - K_d \nabla^2 z \tag{6}$$

where U is the uplift rate [myr$^{-1}$].

A DEM raster grid of the present-day Harir Anticline and the surrounding basins (Fig. 6a) served as model input. The advantage of using Harir Anticline was that the evolved drainage network overprinted the pre-existing one. The boundary conditions were set as closed on all sides except in pre-existing outlets in the input grid. The basins surrounding Harir Anticline were also included in the input grid to minimize the effect of the boundary conditions on the Harir Anticline itself. In the input raster grid, a flow route of each cell was connected with neighbouring cells both diagonally and orthogonally. This means that each cell had the possibility to be linked with eight surrounding cells across its sides and corners (Hobley et al., 2017; Tucker et al., 2016).

Concerning the parameter used in the model, the value of m/n, n, and K were found following the methodology described in (Harel et al., 2016; Mudd et al., 2014; Perron and Royden, 2013), and by comparison with data from Harel et al. (2016). The value of m/n was found by plotting the elevation against X (elevation-X plot) for streams in the input grid (Fig. 6a), where X is found following the equation described by (Perron and Royden, 2013):

$$X = \int_{xb}^{x} \left(\frac{A_0}{A(x)}\right)^{m/n} dx \tag{7}$$

where $A_0$ is the reference drainage area [m$^2$] of 166160 m$^2$ and x is the horizontal upstream distance [m]. In this approach, we ascribed values for m/n range from zero to one, and X was calculated for each time step from Eq. 7. The value of m/n with



maximum regression ($R^2$) value in the elevation-X plot was taken as the best-fitting value, which was 0.41 in our case for the present-day Harir Anticline's drainages (Fig. 6b). This value of m/n is located within the theoretically predicted values of m/n, which ranges from 0.3 to 0.7, from the stream power incision model (Kwang and Parker, 2017; Temme et al., 2017; Whipple and Tucker, 1999).

In the model, $n = 1.7$ and $K = 3.0E\text{-}6$ $yr^{-1}m^{-0.4}$ were used; these values were estimated as mean of K and n in Harel et al. (2016) for those areas that are comparable with our study area in aspect of lithology, climate, and precipitation. The value of m was 0.7. We used an incision threshold of $C = 1.0E\text{-}5myr^{-1}$, which is widely adopted for erosion of an upland landscape (Hobley et al., 2017). The current elevation of the Bekhme and Aqra formations in the crest of Harir Anticline is about 1500 m above sea level. Above that, 2072 m of Upper Cretaceous-Miocene units (Law et al., 2014) and 300 m of Late Miocene Lower

Bakhtiari were exhumed before exposure of the Bekhme and Aqra formations. If we consider that the Lower Bakhtiari have been deposited close to sea level before onset of the MFF c. 5 Ma, there would be 3872 m of rock uplift at a rate of ~0.00077 $myr^{-1}$, which was used in the model. Since soil (regolith) is rare and very thin when present on the slopes, a low diffusivity coefficient of $K_d = 0.001$ $m^2yr^{-1}$ was used (Fernandes and Dietrich, 1997).

There are minor variations in lithology of the three anticlines (they consist of a thick pile of Late Cretaceous carbonate) and

no variation in climate can be expected in such a relatively local scale. Therefore, no significant variances are expected in the used parameters. Lastly, the parameters were calibrated by comparing the nature of the evolved landscape to other anticlines within the High Folded Zone that are cored by the Cretaceous carbonates and more mature than the Harir Anticline to evaluate how realistic the evolved landscape is.

## 4 Results

### 4.1 Geomorphic Indices

The three studied anticlines are composed of Late Cretaceous carbonate ridges arising in their crests, where the Tertiary clastic rocks have been denuded and now compose the sedimentary filling in the adjacent synclines. The three anticlines are dissected by rivers that form water gaps across them. Bekhme and Zinta gorges cut the Perat and Akre anticlines, respectively. We also observed wind gaps, such as those in the NW end of Harir Anticline (Zebari and Burberry, 2015). Therefore, neither the

location of these water and wind gaps nor the plunging tips of anticlines have been considered in interpreting the geomorphic indices as proxies for relative landscape maturity.

The anticlines reach up to c. 1500 m asl, the minimum altitude is c. 400 m in the Greater Zab river course and c. 700 in the adjacent synclines. The hypsometric curves for the three anticlines are presented in Fig. 7. Harir Anticline's curve is more convex, and its shape is close to the youthful stage of Strahler's diagram (Ohmori, 1993; Strahler, 1952) with 68% of the area

above the mean elevation, while Akre Anticline is less convex and close to a mature stage with only 39% of the area above



the mean elevation. Perat Anticline's values are located in between and closer to the Harir Anticline curve with 60% of its area above the mean elevation.

The next three calculated geomorphic indices seem to be substantially influenced by the local structure and wind and water gaps (Fig. 8). Hypsometric integral values vary between 0.2 and 0.77, with lower values in the adjacent synclines and higher

values in the crest of the anticlines. The HI values decrease toward the plunging ends of the anticlines and at water gaps, e.g. Perat Anticline's HI values are maximum at the Greater Zab River. In general, Harir Anticline shows higher values (≤ 0.77) than the other two anticlines. Harir Anticline has a broad crest and has been incised by narrow valleys. This makes the mean elevation within the moving window in the calculation close to the maximum elevation and, thus, causes higher values of the hypsometric integral. Perat Anticline shows values of ≤ 0.66 on its crest to the west of Bekhme Gorge. Among the three

anticlines, Akre Anticline shows the lowest values of ≤ 0.56 to the east of the Zinta Gorge where it links with the Perat Anticline. In its central part the HI values are ≤ 0.51, which is due to presence of more incised and wider valleys which cause the mean elevation within the window to fall.

The surface roughness values range between 1 and 1.33 in the area. The lowest values of the SR are also present in the adjacent synclines and in the plunging tips of the anticlines. The highest values are associated with the location of water gaps. These

are areas where rivers deeply incised both at Bekhme (≤1.33) and Zinta (≤1.32) gorges. Harir Anticline has lowest surface roughness values of ≤1.14 in the SE. They decrease to ≥1.03 in the central part and to ≥1.04 in the NW. Perat Anticline shows the highest value of ≤1.21 especially in its northern limb. Akre Anticline has SR values ≤1.16 in its central segment and ≤1.19 in a wind gap at the western side.

The results of the surface index range between -0.04 to 0.70 in the three anticlines studied. Few locations show negative values.

These are associated partly with adjacent synclines and with Bekhme Gorge (≥-0.04). Apart from these locations, the area shows positive surface indices. Harir Anticline exhibits higher values on its broad crest (≤0.70). SI values reach ≥0.49 on the crest of Perat Anticline to the east of Bekhme Gorge and ≤ 0.58 to the west of the gorge. The values reach to ≤ 0.54 on the crest of Akre Anticline east of Zinta Gorge and ≤ 0.38 east of the gorge. These high values of SI highlight the flat areas with high elevation and high hypsometric integral. The surface index values also highlight the Pila Spi and Khurmala limestone

ridges encircling the anticlines with values close to zero.

The geomorphic indices vary depending on the resolution of the DEM and the size of the moving window (Andreani et al., 2014; Obaid and Allen, 2017). Andreani et al. (2014) found that the DEM resolution does not affect the hypsometric integral, but it affects the surface roughness, while the size of the moving window affects both hypsometric integral and surface roughness. The results become smoother with increasing size of the moving window. In our case, the results also change with

changing the size of the moving window and resolution of the input data (see supplementary material). Since our main aim is to constrain relative maturity levels along the studied anticlines with 6-7 km width, it is reasonable to use a 100x100 cell (3x3 km) moving window which covers approximately a limb of anticline each time and therefore highlights the desirable signal. A smaller window resolves smaller local features rather than the anticlines as a whole, and a larger window does not resolve the main anticlines themselves.



Based on these geomorphic indices of the three anticlines we conclude that there is a measurable difference in landscape maturity between them. The difference in the maturity level must be due to a difference in one or more of the factors tectonics, climate, or rock erodibility. No variation in the climate is expected along the scale of these anticlines, therefore its impact on the landscape maturity can be neglected. The three anticlines show essentially the same lithology (Figs. 2 and 3). Thus, the

only factors that may vary along the anticlines are uplift rate or onset of the uplift. This can be interpreted with one of the following scenarios: either the anticlines started to uplift in the order (1) Akre, (2) Perat and (3) Harir from west to east, or all of them started at the same time but with different rates. In the latter case, the uplift rate would have been highest at Akre and lowest at Harir.

## 4.2 Landscape Model

The aged landscape from the model run is the result of fluvial erosion and hillslope diffusion on the one hand, and uplift due to folding on the other hand. In the landscape modelling, various simulations with different parameters and time spans were performed. Harir Anticline was used as an input model and the landscape evolution model was run for a time span of 10 kyr up to 100 kyr and then it was run for a time span of 20 kyr. The evolving drainage system overprints the pre-existing one in the input and gradually becomes more deeply incised from the anticline flanks curving toward its core (Fig. 9). Harir Anticline

is a box-shaped anticline with a wide and plain crest area. With ongoing incision towards the core of the anticline, this plain crest narrowed gradually and finally became a sharp ridge that divided the drainage basins on the SW flank from those in the NE.

We compared the hypsometric curves of the model outputs to the present-day curves of the anticlines (Fig. 10). Statistically, the hypsometric curve of Harir Anticline was closest to the present-day Perat Anticline after 70 kyr of erosion. The output

curve after 200 kyr matched best with present-day Akre Anticline (with minimum RMS). We conclude that it will take Harir Anticline about 70 kyr to reach the maturity level of Perat Anticline and 200 kyr to reach the level of Akre Anticline if the uplift rates of the three anticlines were the same. The other possibility is that the anticlines started to grow at the same time but with different uplift rates. In this case, it is not possible to find the difference in uplift rates via our landscape modelling. Since the factors that control geomorphology (lithology, structural setting, and climate) were similar for all three anticlines,

and under the assumption of constant growth and erosion rates, we infer that uplift of Akre an Perat anticlines started respectively 200±20 kyr and 70 kyr before Harir started to grow if their uplift rates were the same.

## 5 Discussion

### 5.1 Rock Erodibility

As described in section 2.2 and Fig. 3, the stratigraphic column in the area consists of rocks with different erodibility. In

general, the Cretaceous carbonate units of Qamchuqa, Bekhme, and Aqra formations and the Paleogene carbonates of Sinjar,



Khurmala, and Pila Spi formations are more resistant to erosion and form outstanding ridges. The Upper Cretaceous-Tertiary clastic rocks are less resistant to erosion. The latter units have been eroded to the ground level where they crop out, except in some areas where they form a badland landscape (e.g. Bakhtiari group). The progress of erosion along with the uplift due to folding can be separated into several stages that reflect the resistance of the exposed rock units to erosion (Figs. 2 and 11):

first, the area is covered by Quaternary sediments in unfolded, very wide synclines in between anticlines, especially in the Foothill Zone (e.g. in Akre Plain to the south of Akre Anticline). Next, the Neogene Fars and Bakhtiari units expose in the next stage of folding and produce badland landforms without a prominent relief. Their surrounding areas do not exceed 500 m asl (e.g. in Sarta Anticline; Fig. 11). Then, Paleogene carbonates expose in the core of the anticlines and form outstanding whale-shaped anticlinal ridges with relief exceeding 750 m (e.g. Pirmam anticline; Figs. 2 and 11). In the next erosional step,

the Cretaceous carbonates expose in the core of anticlines and form anticlinal ridge also with relief exceeding 2 km (e.g. many anticlines within the High Folded Zone including the three studied anticlines). Finally, the Upper Triassic-Lower Cretaceous evaporites, shale, and bedded limestone units are exposed where the Cretaceous carbonates have been eroded down to the cores of the anticlines, especially in those that are located to the north and northeast close to the High Zagros Fault.

Currently, the studied anticlines are in the stage in which the Cretaceous carbonates form the main anticline body. The maturity

level along these anticlines therefore represents the level when these carbonates cropped out in their latest stage.

## 5.2 Landscape Maturity and Modelling

Any relative change in the base level induced by tectonics or climate leads to the change of erosion rates. A landscape survives when its uplift is not completely counterbalanced by erosion (Andreani and Gloaguen, 2016; Burbank and Anderson, 2012; Pérez-Peña et al., 2015). The relative relict landscape and its distribution on these three anticlines atop the MFF reveal clues

about underlying tectonics since there is no significant variation in climate and lithology.

Within the three studied anticlines, the geomorphic indices effectively highlighted their incision. The locations dissected by rivers show high surface roughness. The surface index which combines both hypsometric integral and surface roughness, sets apart relict landscapes of positive and high values from transient landscapes of negative values that are preferentially incised (Andreani et al., 2014). There is a notable relative declination in areas where anticlines are crossed by rivers e.g. Bekhme and

Zinta gorges, which show a high surface roughness. Also, variations in surface index are found in comparable areas in the three anticlines. Focussing on the crest of the anticlines, we see that Harir Anticline shows higher values than the two others. The lowest values are found in Akre Anticline. Harir has low incision at elevated surfaces while Akre has more incised uneven landscapes and the erosion waves have functioned deeper into the core of the anticline. This can also be seen from the valley shapes. We observe a narrow V-shape in the flanks of Harir and a wide V-shape in Akre (Figs. 12a and 12b). The same effect

is visible in swath topographic profiles (Figs. 12c and 12d): in Harir Anticline, there is a clear topographic step with a higher slope angle, while in Akre Anticline the slope is gentler and more linear. We relate this difference to the underlying tectonics. This can be interpreted with one of these premises: either both anticlines started to uplift successively (first Akre, then Perat,



and finally Harir), or all of them started at the same time but with different uplift and exhumation rates (Akre the fastest, Harir the slowest).

The current landscape of these anticlines exposes Cretaceous carbonates of the Qamchuqa, Bekhme and Aqra formations, which became exposed to erosion only after unroofing of the entire Palaeogene to Neogene succession. The youngest

stratigraphic unit affected by folding is the Upper Miocene-Pliocene Bakhtiari group, as observed from growth strata (Csontos et al., 2012). This has also been observed in the Upper Bakhtiari (Pliocene-Pleistocene) close to the MFF (Koshnaw et al., 2017). In between Bekhme and Aqra and the Upper Bakhtiari formations, 2.37 km of the Upper Cretaceous to Miocene clastic rocks interbedded with thin units of limestone (Law et al., 2014) have been exhumed due to successive rock uplift in the crest of the studied anticlines, triggered by shortening and erosion. They are only preserved in the adjacent synclines. The Cretaceous

carbonates themselves have been exposed in the crests of Akre, Perat, and Harir anticlines for c. 0.9 km above the level of the other exhumed units. Based on the thickness, the amount of the exposed Upper Cretaceous carbonate makes c. 28 % of the total exhumed and exposed thickness in the crest of the anticlines. Therefore, with both scenarios (different uplift time or different uplift rate) and with assumption of constant (linear) rock uplift rate through time, the Upper Cretaceous carbonate in Harir Anticline was exposed to erosion later than in the Akre Anticline (Fig. 13a).

The steeper valley flanks in Harir Anticline compared to those of Akre also support higher uplift rates of the Harir Anticline. Furthermore, the relationship between slope and drainage area for streams in the Harir Anticline is positive (Fig. 13b), which means the streams have a convex shape and the streams' segments with steeper slopes are still located in the flanks of the anticline. In the Akre Anticline, this relationship is negative (Fig. 13b), which means that the streams have a concave shape and the segments with steeper slopes have migrated toward the core of the anticline. This implies that tectonic activity in the

Harir Anticline is younger than in the Akre Anticline. Therefore, the premise of having Harir Anticline starting its uplift later than Akre Anticline is most likely.

Since the Upper Cretaceous carbonates in Harir Anticline were exposed later than in Akre Anticline, a landscape evolution model is a viable approach to estimate the exposure time difference. Here the model is built for the first premise of different onsets of uplift. Even if the second premise of different uplift rates is correct, the estimated time difference of the carbonate

exposure will only be 28% less than that for the first scenario. As described in section 4.2, the calculated uplift time difference between Akre and Harir anticline is 200±20 kyr, and if the second scenario is correct, the time difference of the carbonate exposure would be 144±14.4 kyr.

In the model, various parameters and two well-known landscape evolution equations were used, but in general it is impossible to mimic nature perfectly. The time variation was measured assuming that the climate did not change much during the evolution

of the landscape due to lack of paleoclimate data and for the sake of simplicity, acknowledging that climatic change has a significant impact on the landscape. In addition, neither rock fall nor karstification were included in the model for simplicity. Field observations suggest that karstification does not have a significant impact on the landscape. Overall, the evolved landscape from the model seems to be plausible in comparison with the other anticlines that surround Harir Anticline, and the landscape models are more mature with respect to topography that developed and the overall drainage patterns.



## 5.3 Structural Style and regional tectonics

We constructed structural cross-sections for the three anticlines from field data and literature (Fig. 4; Syan, 2014). These cross-sections show thrust-related folds in accordance with published studies of the area (Csontos et al. 2012). The anticlines are box-shaped with broad hinge zones and doubly plunging. All three anticlines show a thrust fault in their forelimb. The Perat

Anticline has a thrust in its backlimb, as well. The anticlines have a steeper forelimb which is nearly vertical, and the strata become overturned within the Pila Spi, Sinjar, and Khurmala formations. Using line-length balancing, the shortening of the Upper Cretaceous strata is measured from the constructed cross-sections to be 26%, 31%, and 29% in the Harir, Perat and Akre anticlines, respectively. The cross-sections show variations in the stratigraphic thickness in between the three anticlines, especially in the thickness of the Late Cretaceous-Tertiary clastic rocks. The clastic rocks are less resistant to erosion and the

raised body of the anticlines is entirely made up of Cretaceous carbonates. Therefore, it is not expected that these variations in the structural geometry and stratigraphic thickness have much impact on the variation of the landscape maturity in the three anticlines.

An orogenic bend is depicted in the area where the trend of structures changes across the Greater Zab River from NW-SE at the eastern side of the river to nearly E-W at its western side. The course of the Greater Zab River is suggested to overlie a

NE-trending transversal basement fault with right-lateral displacement (Ameen, 1992; Burberry, 2015; Jassim and Goff, 2006; Omar, 2005) and there is an offset of the High Folded Zone propagation foreland-ward. At the eastern side of the river the deformation has propagated for about 25 km further than on its western side (Figs. 1 and 2). The origin of this fault reaches back to the Late Proterozoic tectonic history of the Arabian Plate, and the fault has been reactivated later in subsequent tectonic events (Ameen, 1992; Aqrawi et al., 2010; Burberry, 2015; Jassim and Goff, 2006). This can also be noticed in the thickness

of the sedimentary cover, which is thinner to the west of the Greater Zab River (Ameen, 1992; Zebari and Burberry, 2015). This is most likely due to a series of uplift events and erosional/non-depositional gaps during the Mesozoic (Ameen, 1992; Aqrawi et al., 2010), which in turn may have influences the foreland-ward propagation of deformation (Marshak and Wilkerson, 1992). The deference in propagation of deformation may also be due to rotation of the belt trend as the convergence direction changed (Csontos et al., 2012) from NW-SE in the eastern side where the convergence is accommodated by belt-

normal slip and right-lateral strike-slip to nearly E-W in the west where the convergence is mostly accommodated by belt-normal slip (Reilinger et al., 2006).

Zebari and Burberry (2015) found that anticlines to the east of the Greater Zab River (Harir, Shakrok and Safin anticlines) demonstrate pronounced NW-ward propagation based on their geomorphic criteria, and the start point of the NW-ward propagation of the Harir Anticline is close to its SE end (their Figs 16 and 20). This implies that progressing uplift in the

hanging wall of the MFF was not gradually continuing from the Akre Anticline towards the Perat Anticline and further SE-ward to the Harir Anticline. The uplift progress is probably rather partitioned into segments along the belt. In addition, other anticlines to the south (Safin Anticline) and to the southwest (Shakrok Anticline) of Harir Anticline are more mature than Harir Anticline itself based on their hypsometric curves (Fig. 14) and geomorphic indices (supplementary material), implying that



the foreland-ward propagation of the deformation was also out-of-sequence in this part of the High Folded Zone. This has also been noted in the Foothill Zone based on thermochronological dating (Koshnaw et al., 2017) and landscape maturity (Obaid and Allen, 2017). Thus, the most plausible scenario is that deformation in the Harir segment started sometime after that in Akre segment (200 kyr according to our landscape evolution modelling). Harir Anticline uplift would also postdate Perat

Anticline uplift (70 kyr) to the west and the onset of Safin and Shakrok anticlines to the south and southeast, which are not included in the model (Fig. 15). As discussed by Csontos et al., (2012), the fold relay corresponds to the change in strain partitioning and rotation of the horizontal stress direction from the NE-SW to N-S in Late Pliocene (Navabpour et al., 2008). During the latest stage of the N-S convergence, a right-lateral shear and superposed folding along the NW-SE trending anticlines (Csontos et al., 2012) can be observed from the relay of the Shakrok, Harir and Perat anticlines (Figs 2 and 15).

Applying this concept requires a comprehensive paleostress analysis investigation especially within these studied anticlines, which is beyond the scope of this paper.

## 6 Conclusions

The geomorphic indices used in this study allowed us to quantitatively differentiate between variably degraded landforms in the frontal Zagros Mountain of NE Iraq. This area is characterized by active folding due to ongoing convergence between the

Eurasian and Arabian plates. Three active thrust-related anticlines that are aligned along-strike the MFF were studied in detail. While the Akre Anticline shows deeply incised valleys indicative of advanced erosion, the Harir and Perat anticlines have relatively smooth surfaces and are younger than Akre. We related this difference to the underlying tectonics. This can be interpreted with one of the following concepts: either anticlinal growth started at different times or all of them started to grow at the same time, but with different surface uplift and exhumation rates.

A comparison of the geomorphic indices values of the model output with those of the present-day Akre Anticline topography revealed that it will take Harir Anticline 200±20 kyr to reach the maturity level of today's Akre Anticline and 70±10 kyr to reach the maturity level of the Perat Anticline assuming constant uplift rates along the three anticlines. Due to similarity in the lithology, structural setting and climate along the three anticlines and by assuming constant growth and erosion rates, we infer that Akre Anticline started to grow 200±20 kyr before Harir Anticline. The onset of growth of Perat Anticline lies closer to

that of Harir. A NW-ward propagation of Harir Anticline itself implies that the uplift has been independent within different segments rather than having been continuous from the NW to the SE. Our method of estimating relative age differences in variously degraded anticlines can be applied to many other anticlines along the MFF and could eventually develop into a model of the temporal evolution of this fold and thrust belt.



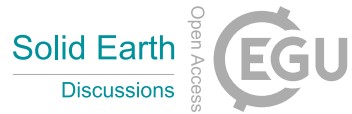

**7 Acknowledgments**

The German Academic Exchange Service (DAAD) is acknowledged for providing a scholarship (Research Grants – Doctoral Programmes in Germany, 2016/17; 57214224) to the first author to conduct this PhD research in Germany. The authors express their gratitude to the German Research Foundation (DFG) project no. 393274947 for providing financial support. The German

Aerospace Agency (DLR) and Mr. Thomas Busche in particular are thanked for providing TanDEM-X digital elevation models.

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





**Figure 1: Tectonic subdivision of the NW segment of the Zagros Fold-Thrust Belt (modified after Berberian, 1995; Emre et al., 2013; Koshnaw et al., 2017; Zebari and Burberry, 2015).**





**Figure 2: Geological map of the Zagros belt in KRI showing the location of the three anticlines Harir, Perat, and Akre with respect to the MFF that separates the High Folded Zone from the Foothill Zone (modified after Csontos et al., 2012; Sissakian, 1997; Zebari and Burberry, 2015).**





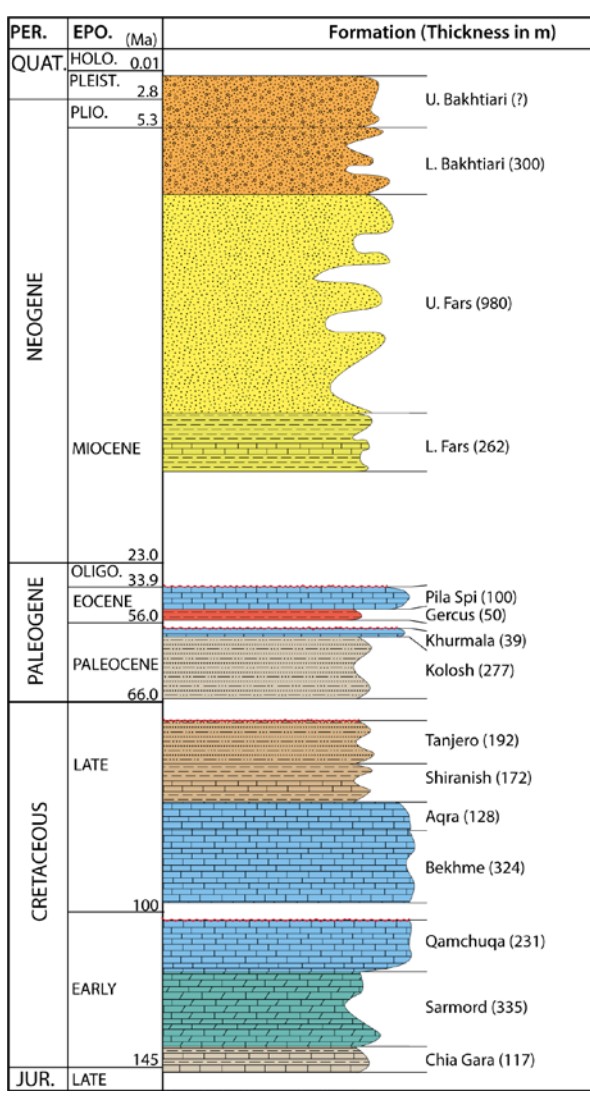

**Figure 3: Stratigraphic column of the exposed rock units in the area. Thicknesses are given as in well Bijeel-1 (Fig. 2), which is located 5 km to the south of Perat Anticline modified after (Law et al., 2014). The column is scaled to the stratigraphic thicknesses.**





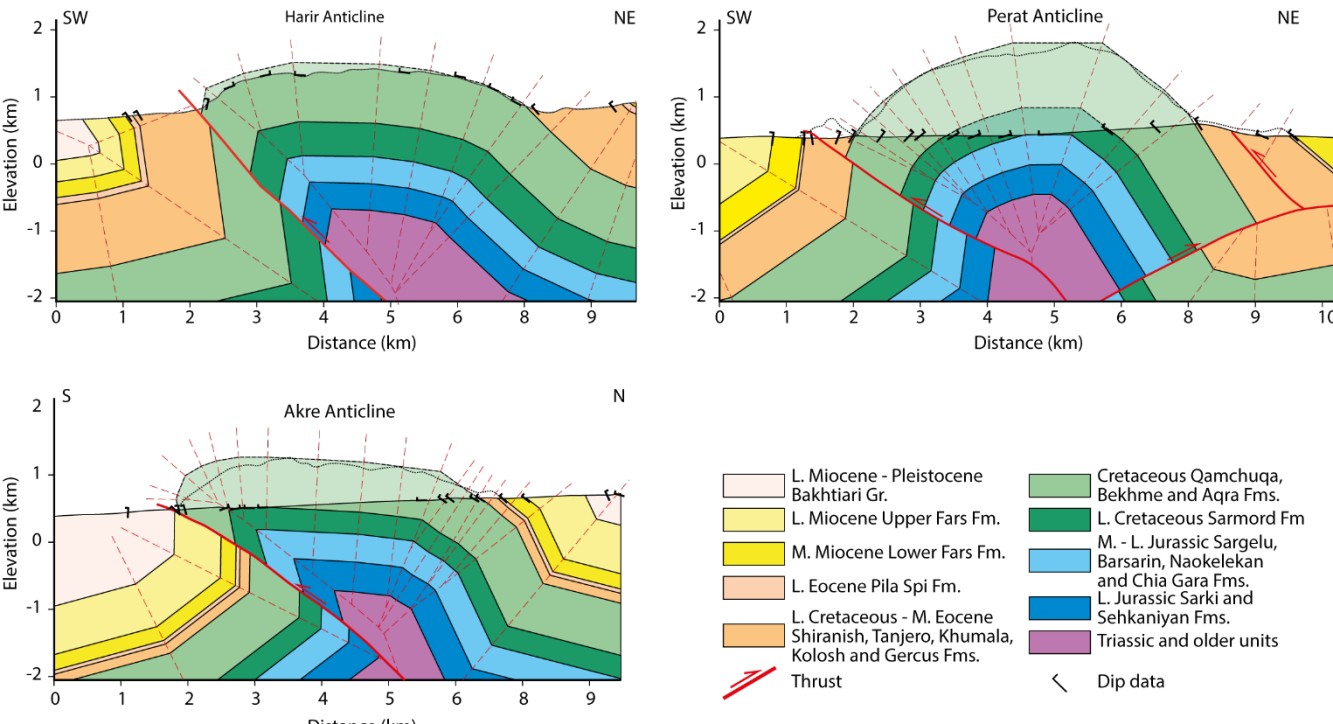

**Figure 4: Structural cross-section across the three studied anticlines; a) Harir section (modified after Syan, 2014), b) Perat section constructed from field data and thrusts inferred from an interpreted seismic line by Csontos et al. (2012), c) Akre section constructed from field data (see Figure 2 for the locations).**





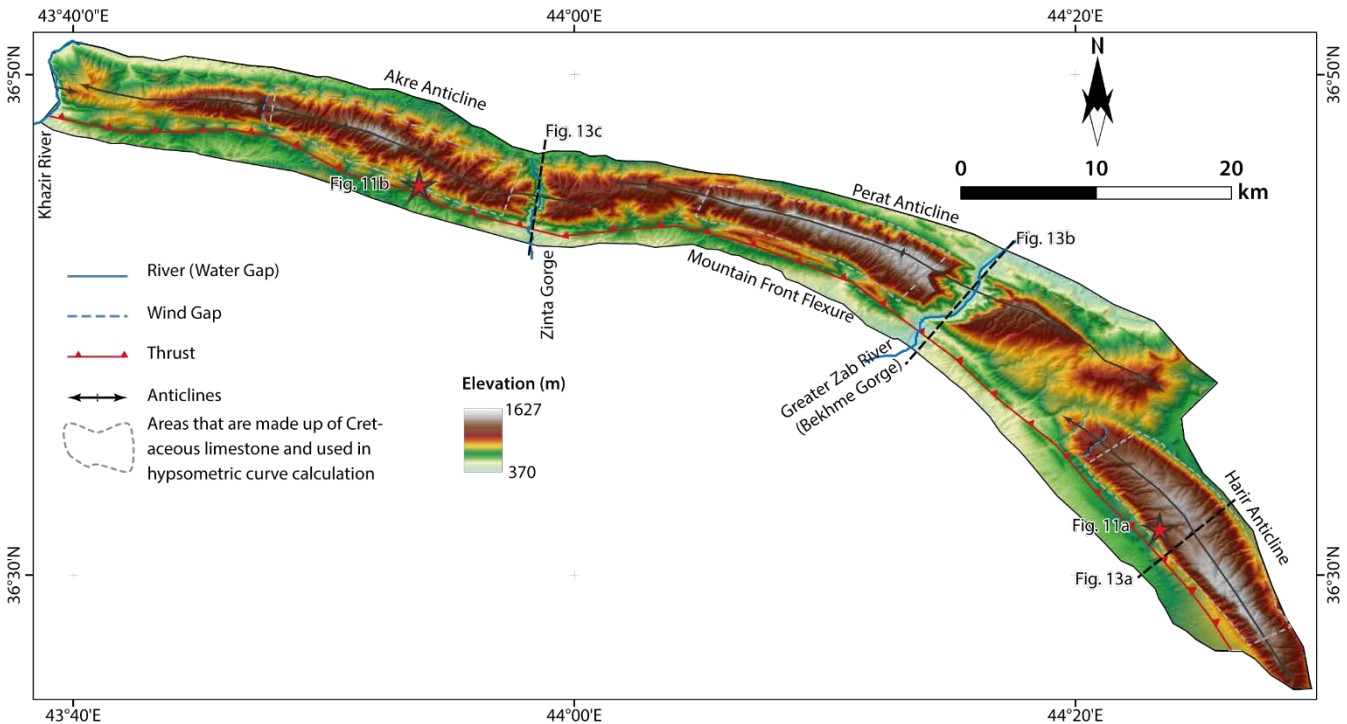

**Figure 5: Topography of the studied anticlines obtained from 30 m resolution SRTM1 DEM data showing the location of water and wind gaps across these anticlines.**

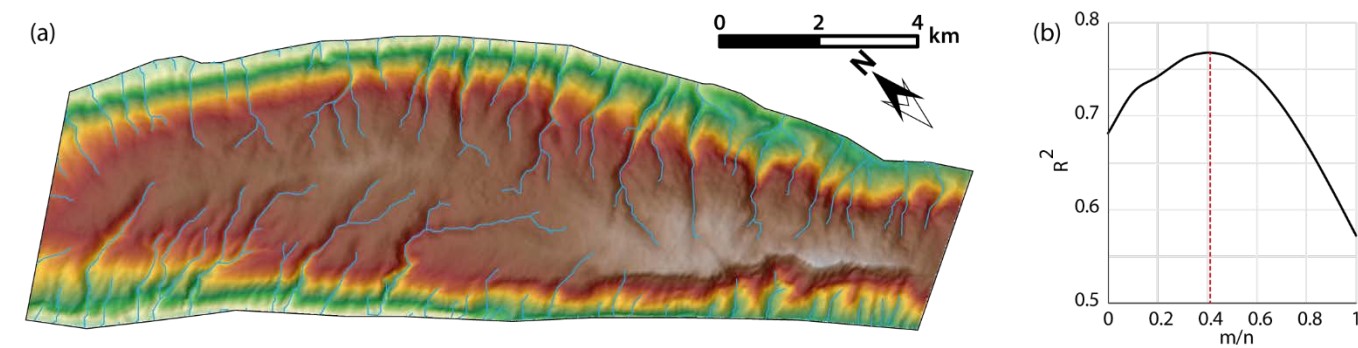

**Figure 6: a) DEM grid and drainage network for the present-day Harir Anticline that is used as an input for the model; b) m/n plotted against regression values of elevation-X plot for streams in the Harir Anticline. The highest regression is achieved for m/n = 0.41.**





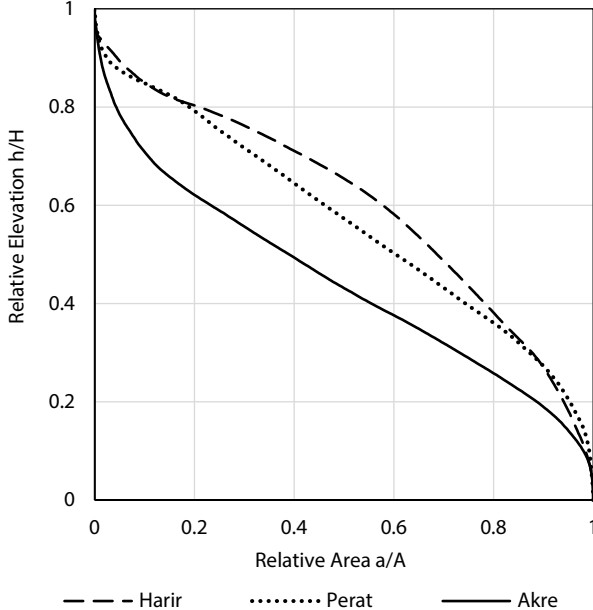

**Figure 7: Present-day Hypsometric curves of the studied anticlines. We only use those parts where Upper Cretaceous carbonate rocks crop out and we exclude wind gaps, water gaps, and the plunging tips of anticlines.**

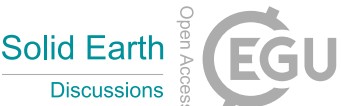

**Figure 8: Surface index maps for the three anticlines calculated from 100 x 100 pixel cells (3 x 3 km) and moving windows; a) hypsometric integral, b) surface roughness, and c) surface index.**




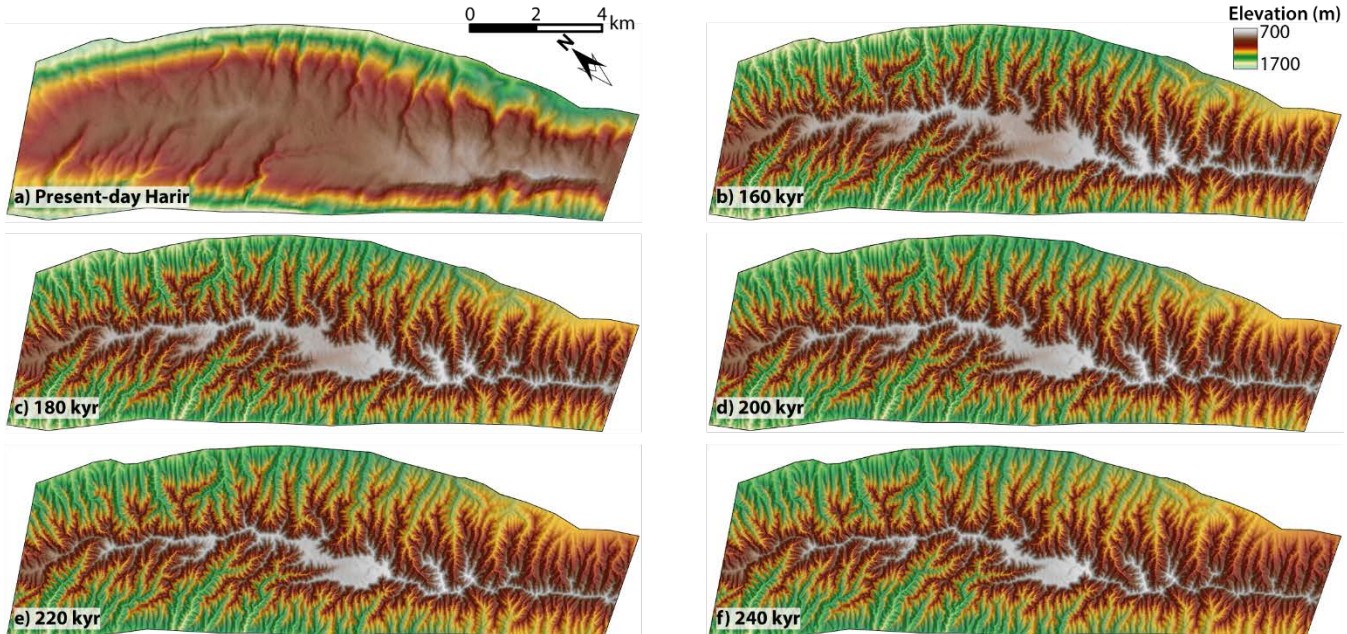

**Figure 9: The input landscape (a), which is present-day Harir topography, and the evolved landscape through time; b) 160 kyr, c) 180 kyr, d) 200 kyr, e) 220 kyr, and f) 240 kyr.**

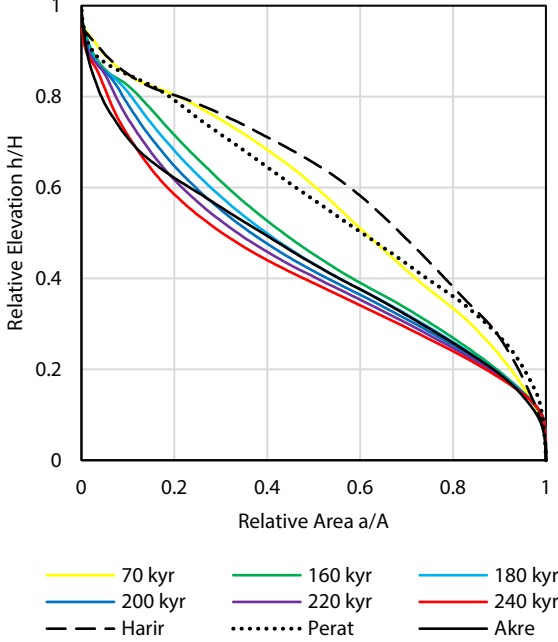

**Figure 10: Hypsometric curves of the studied anticlines and those of the evolved Harir landscape from the model for six different time spans.**



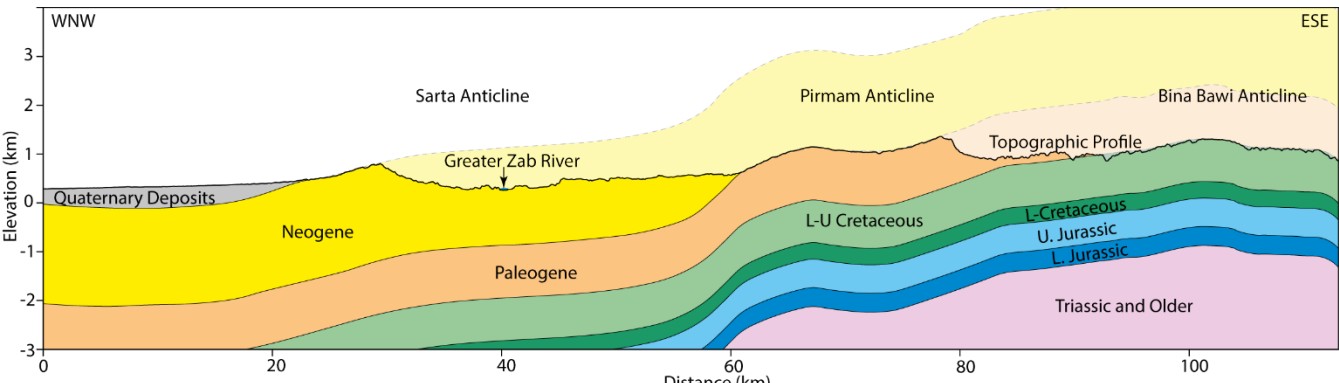

**Figure 11: Topographic profile along the axis of Bina Bawi, Pirman, and Sarta Anticlines (see Figure 2 for location), showing the distinctive successive stages of uplift/erosion due to folding within the Zagros Fold-Thrust Belt in KRI.**

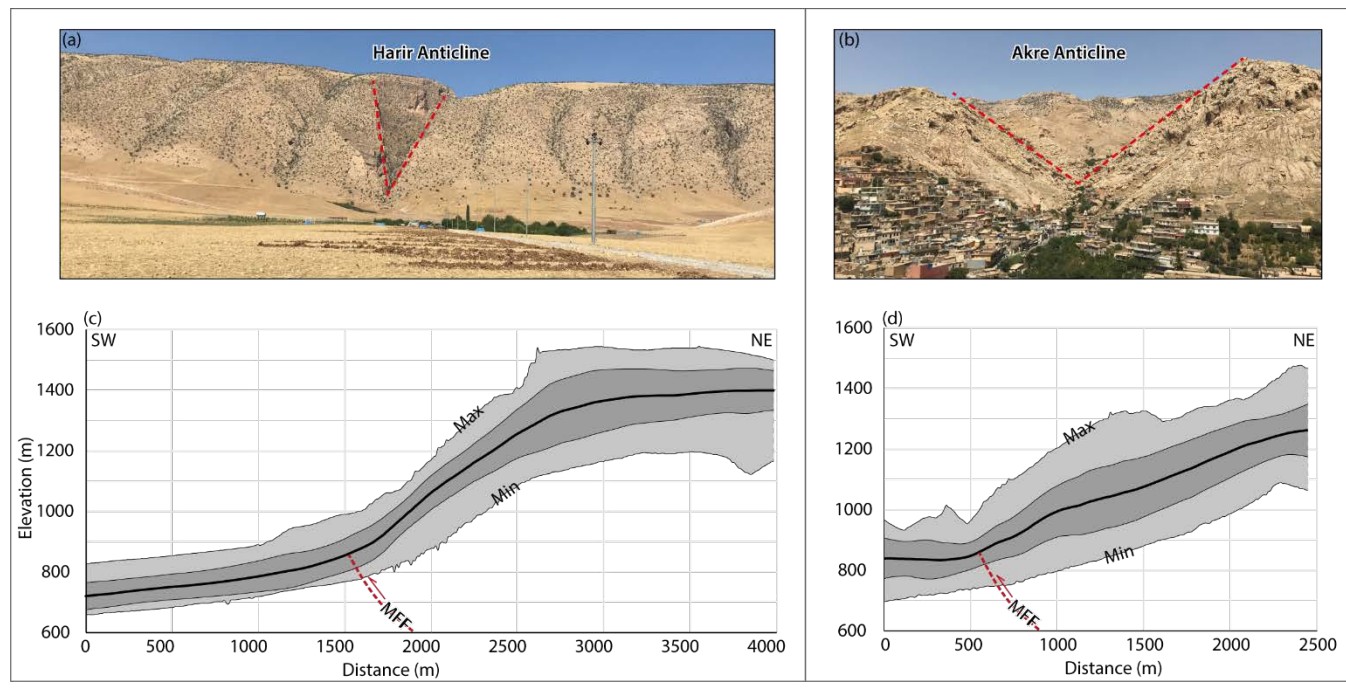

**Figure 12: Different shape of valleys in Harir (a) and Akre Anticlines (b; see Figures 2 and 5 for the locations) and swath topographic profiles across the southern limb of Harir (c) and Akre (d) Anticlines. Right side of the topographic profiles mark the locations where the Pila Spi Fm crops out in the anticlines' crest. MFF: Mountain Front Flexure.**



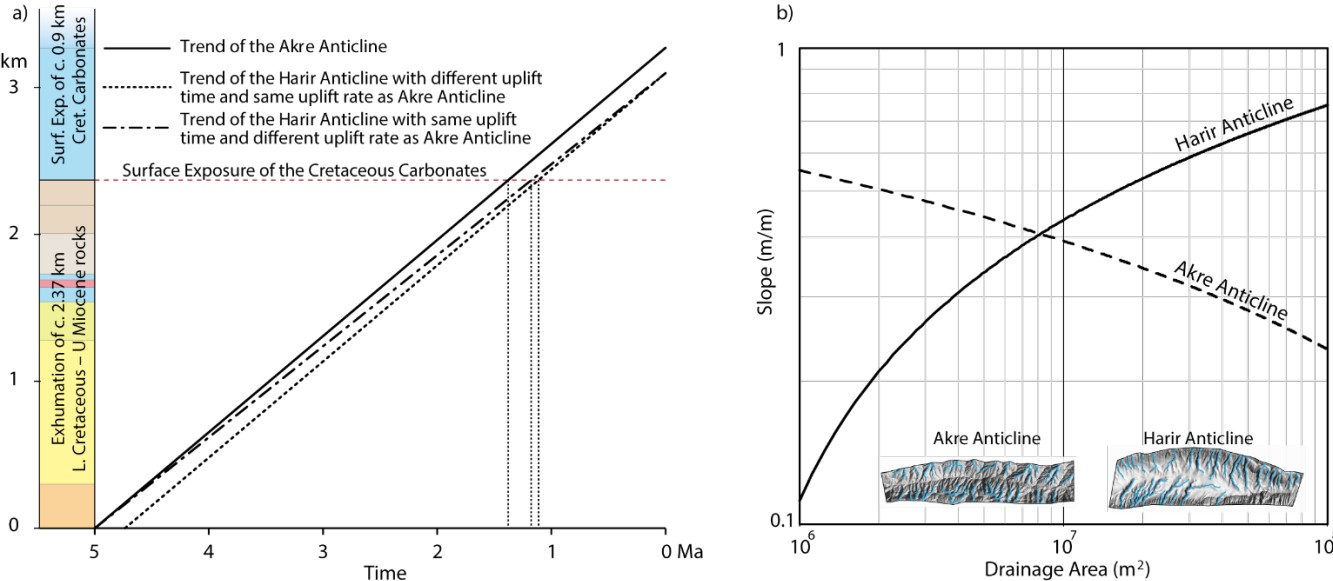

**Figure 13: a) Diagram showing the exposure time of the Upper Cretaceous carbonates in Akre and Harir Anticlines. Two different scenarios are plotted for Harir: Having a slower uplift rate than Akre, or onset of uplift later than Akre. b) Channel slope-drainage area plots of streams in both Akre and Harir Anticlines.**

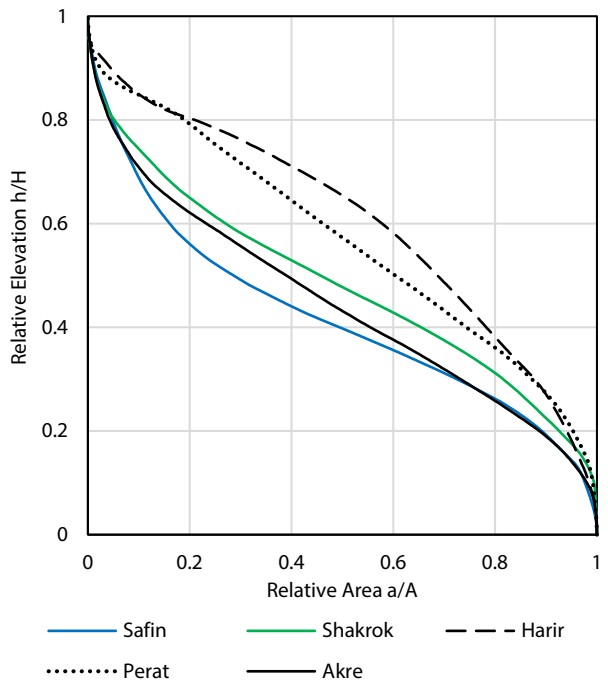

**Figure 14: Hypsometric curves for the studied anticlines as compared to those of the Shakrok and Safin anticlines, which show that the Harir's curve is more convex than that of both Shakrok and Safin.**

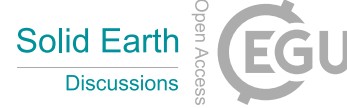

**Figure 15: Simplified history of the formation of anticlines during the propagation of the deformation front over time in the study area. The Harir anticline is likely the latest to have formed within the High Folded Zone in its SE end. It occupies the position of a relay structure during the linkage of two adjacent, but overlapping segments of the deformation front.**