# Peer review of "Relative Timing of Uplift along the Zagros Mountain Front Flexure (Kurdistan Region of Iraq): Constrained by Geomorphic Indices and Landscape Evolution Modelling"

_Solid Earth, 2018_

## Referee Comment (RC1) · Anonymous Referee #1 · 27 Dec 2018

The authors of this manuscript try to use geomorphic indices and results of landscape modelling to constrain the relative timing of uplift of three anticlines. In general, the topic is interesting and it will be a substantial contribution to the journal. Nevertheless, the revisions including the methodology and discussion, as well as the rearrangements of sections are still needed before publication. Major comments and suggestions are listed below.

1. Introduction: the authors should clearly state the importance of this study. Why the detailed spatial and temporal distribution of deformation ... is not yet well understood ?

[Figure]

Due to the lack of subsurface data, and/or this region is inaccessible for field surveys?

2. Section 3.1.1: with aim of assessing landscape maturity along thrust-related anticlines, hypsometric curves and integrals have often been used for (sub-) drainage basins. The methodology differs from the three incomplete hypsometric curves displayed in Fig. 7. Actually, the authors did not extract drainage basins even if the stream channels of the Harir anticline have been shown in Fig. 6a.

3. Section 3.1.5 Digital elevation models: this section does not belong to the 3.1 geomorphic indices.

4. Section 5.1: the authors just described the rock erodibility. They should be included in geological setting, instead of discussion part. Here, the authors stated " the stratigraphic column in the area consists of rocks with different erodibility" (page 11, line 29), and also mentioned in the conclusion "Due to the similarity in the lithology, structural setting and climate" (page 15, line 23-24). They should clearly state whether the difference exists or not.

---

## Referee Comment (RC2) · Anonymous Referee #2 · 8 Feb 2019

Comments to the manuscript entitled "Relative Timing of Uplift along the Zagros Mountain Front Flexure Constrained by Geomorphic Indices and Landscape Modelling, Kurdistan Region of Iraq" by Zebari et al. (doi:10.5194/se-2018-124).

The authors of this manuscript try to constrain the relative timing of uplift of three anticlinal folds of the Iraqi Zagros Mts., combining the results of landscape evolution models and geomorphic indices. The topic fits the ones of the journal and the manuscript has the potential to be interesting for the international scientific community. Nonetheless, some general comments and minor specific ones are listed below, suggesting that

[Figure]

some important revisions are needed before publication.

general comments 1) Considering the deformation style of the Folded Zone of the Zagros Mts. chain, the assumption of constant rock uplift seems too simplistic. Doesn't the evidence of NW-ward propagation of the Harir Anticline (used by the authors for supporting the scenario of independent and diachronist uplift in different fold segments) affect the assumption of homogeneous and constant uplift rate in each structure? In this frame, the hypsometric analysis performed for the entire anticlinal ridges seems to have no sense, while I would suggest to use an approach to the hypsometric analysis such as the one proposed by Pérez-Peña et al. (2009). 2) Also the use of the only equations for fluvial erosion and diffusion processes for landscape evolution modelling may be too simplistic. Besides the justifications provided by the authors in section 5.2, it sounds not realistic that sedimentation on the slopes of anticlines can be neglected over the time-span of landscape evolution modelling (105 years), as well as the assumption of constant erosion rates (and climate!). 3) The authors justify the choice of the present topography of the Harir Anticline as LEM input asserting that in this structure the evolved drainage network overprinted the pre-existing one. Looking at Fig. 6 it seems that the drainage has a pattern similar to the one described by Ramsey et al. (2008; Basin Research (2008) 20, 23–48, doi: 10.1111/j.1365-2117.2007.00342.x) as evidence of lateral propagation of folds in the Zagros. This implies a diachronic fingerprint in the drainage network which could void the sense of performing the hypsometric analysis for the entire anticlinal ridges. 4) Some of the units stratigraphically above the Cretaceous limestones outcropping on the anticlines' crest are transitional to continental (i.e. the Bakhtiari Fm.), thus likely being affected by lateral variability of thickness. What about the effects on the uplift rate calculation based on thickness? Furthermore, this uplift rate was calculated based on the thickness and elevation of units on the anticline crest, but (again) is it correct to extend such a rate to the entire folds given their lateral growth? 5) There are several repetitions over the manuscript (see "specific comments") 6) Some original data (geological cross sections of Fig. 4) are referred to in the geological setting, while should be better described in the results. 7) In some

cases, the interpretations seem not supported by data. For example, the fit between some of the hypsometric curves obtained with LEM and the ones computed for the three analyzed anticlines is not evident in Fig. 10 and the "minimum RMS" invoked by the authors to demonstrate the fit is not quantified. On the other hand, authors provide a quite specific timing for the inferred "delay" in the deformation sequence of the three folds which is based on this "fit" and use it to support the diachronic scenario of fold development. In my opinion such a constrain is weak, if based on the hypsometric analysis. Other doubt interpretations are listed in the specific comments. 8) Section 5.1 doesn't sound necessary 9) In the Discussion new data are presented (i.e. Fig. 13), but it is not explained how they have been obtained, in particular the calculation of the slope/area. Is it obtained using just the drainage network or the whole topography? 10) In the Conclusions authors refer to the three analyzed anticlines as "active folds", while in section 5.2 they state that the youngest unit affected by folding is the Mio-Pliocene Bakhtiari Fm.

specific comments and technical corrections -TITLE: I would suggest to change the title into: "Relative Timing of Uplift along the Zagros Mountain Front Flexure (Kurdistan Region of Iraq): Constrains by Geomorphic Indices and Landscape Evolution Modelling"

- ABSTRACT: PAGE 1, LINE 13: maybe "fold and thrust belt" and not "fault and thrust belt"

- INTRODUCTION: PAGE 2, LINES 16-17: Why "The timing of this activity is expected to differ along-strike"? Any reference or explanation?

- GEOLOGICAL SETTING: PAGE 4, LINE 31: change "river terraces" into "terraced alluvium". PAGE 5, LINES 18-19: I suggest not to refer to new data in the geological setting. Fig.4 should be described (if made with newly surveyed data) in the Results.

- DATA AND METHODS: PAGE 5, LINE 26: the hypsometric curve is not an "index" PAGE 5, LINE 27-29: the definition/meaning of the geomorphic tools is vague and in some cases, not correct (i.e. "The hypsometric curve and the hypsometric integral

highlight raised and flat surfaces"). PAGE 6, LINES 3-4: in general, the convex vs. concave shape of the hypsometric curve not necessarily reflects the "maturity" of a landscape (in terms of its absolute age), but can also depend on the type and rates of earth surface processes which dominate the landscape evolution (e.g. linear incision vs. hillslope diffusion processes). PAGE 6, LINES 10-11: again, the meaning of HI is not clearly defined. Please, rephrase. PAGE 7, LINE 1: change the order of terms into "Nh, NHI and NSR" PAGE 7, LINE 5: Digital Elevation Models (3.1.5.) are not Geomorphic Indices. This section should become 3.2 PAGE 7, LINE 22: soil creep is mentioned as second main process inputed in LEM. Maybe the authors should refer more generally to hillslope diffusion processes. PAGE 7, LINE 23-24: see general comment 2): it sounds strange that over the time-span of the modelling the sedimentation on slopes can be neglected. PAGE 8, LINE 8: again, the authors refer to "soil creep" (see comment above). PAGE 8, LINES 17-18: see general comment 3). PAGE 8, LINE 30: authors refer to "time steps" before defining them. PAGE 9, LINE 13: "Kd = 0.001 m2yr-1": why exactly this value?

- RESULTS: PAGE 10, LINE 6: "HI values are maximum at the Greater Zab River": maybe authors mean that HI values are minimum? PAGE 10, LINE 6-12: This part seems not necessary and the authors should pay attention to the meaning of HI when calculated for square areas and not for single basins. In this case HI measures how rapidly elevation changes and not strictly the amount of incision. PAGE 10, LINE 13-18: results concerning roughness analysis are quite obvious...is it really necessary? PAGE 10, LINE 26-34: this part should be moved to the methodological section. PAGE 11, LINES 11-12: authors state that "In the landscape modelling, various simulations with different parameters and time spans were performed", nonetheless they do not provide details on the simulation (neither in the supplementary material). How did they select the best outputs? PAGE 11, LINES 13-14: authors state that "The evolving drainage system overprints the pre-existing one in the input and gradually becomes more deeply incised from the anticline flanks curving toward its core (Fig. 9)". This is not evident in Fig. 9, according to what already explained in the general comment 3). PAGE 11,

LINES 15: change "plain" into "flat". PAGE 11, LINES 20-22: see general comment 7).

- DISCUSSION: Is section 5.1 necessary? PAGE 12, LINES 14-15: what the authors mean with "The maturity level along these anticlines therefore represents the level when these carbonates cropped out in their latest stage"? PAGE 12, LINES 17-19: authors state that "A landscape survives when its uplift is not completely counterbalanced by erosion (Andreani and Gloaguen, 2016; Burbank and Anderson, 2012; Pérez-Peña et al., 2015)": it does not sound...maybe authors refer to relict landscapes? PAGE 12, LINES 21-22: the sentence "The locations dissected by rivers show high surface roughness" seems obvious and not necessary. PAGE 12, LINES 29-31: "The same effect is visible in swath topographic profiles (Figs. 12c and 12d): in Harir Anticline, there is a clear topographic step with a higher slope angle, while in Akre Anticline the slope is gentler and more linear": to outline this evidence swath profiles are not necessary...if they can provide further evidence, the latter should be discussed. PAGES 12-13: "This can be interpreted with one of these premises: either both anticlines started to uplift successively (first Akre, then Perat, and finally Harir), or all of them started at the same time but with different uplift and exhumation rates (Akre the fastest, Harir the slowest)". This concept is repeated too many times over the manuscript. Furthermore, to justify the different geomorphic stage of the three folds with different uplift rates, shouldn't the latter be "fastest" in Harir and "slowest" in Akre?? PAGE 13, LINE 13: How much does the assumption of constant rock uplift affect the results obtained? Since it is a "strong" assumption you should give an estimation of that. PAGE 13, LINE 15-26: This part of the discussion is not so clear. âŹč E.g. how did the authors perform the slope/area analysis? âŹč Some statements seem wrong: e.g. "In the Akre Anticline, this relationship [slope/area] is negative (Fig. 13b), which means that the streams have a concave shape and the segments with steeper slopes have migrated toward the core of the anticline. This implies that tectonic activity in the Harir Anticline is younger than in the Akre Anticline. Therefore, the premise of having Harir Anticline starting its uplift later than Akre Anticline is most likely". Why a higher uplift rate in the Harir couldn't have caused the same effect? âŹč "Since the Upper Cretaceous carbonates in Harir Anticline were

exposed later than in Akre Anticline, a landscape evolution model is a viable approach to estimate the exposure time difference. Here the model is built for the first premise of different onsets of uplift. Even if the second premise of different uplift rates is correct, the estimated time difference of the carbonate exposure will only be 28% less than that for the first scenario. As described in section 4.2, the calculated uplift time difference between Akre and Harir anticline is 200±20 kyr, and if the second scenario is correct, the time difference of the carbonate exposure would be 144±14.4 kyr" This sentences are confused and the interpretation is not clear and a bit circular (choice of scenario based on modelling, based on constant uplift rates...). PAGE 13, LINE 27-34: see general comment 2). PAGE 14, LINE 8: The variations in stratigraphic thickness in between the anticlines is constrained by field data? And how does this variability affect the calculations of uplift rates? PAGE 14, LINE 13-26: Is this part really necessary and functional?

- CONCLUSIONS: See general comment 10

FIGURES: FIG.1: i) I suggest to change "transform fault" into "strike-slip fault" in legend; ii) I suggest to add a legend for the different colors in transparency (e.g. the rose one corresponds to 2 zones...); check the use (or not) of parenthesis in the naming of zones; FIG.2: i) I suggest to draw the border between different units in map; ii) check the numbering of the figures recalled in the sketch (they do not correspond to the figure recalled) FIG.4: move and comment it in the Results. FIG.5: symbols for wind gap and limit of Cretaceous limestone outcrop are not visible in figure FIG.6: Symbols for the strike-slip component of displacement are missing FIG.7: i) I suggest to prepare a new figure after having used the approach by Pérez-Peña et al (2009) to the hypsometric analysis. FIGs.8, 9, 10: to be revised according to the comments on the analytical procedures.

---

## Author Comment (AC1) · 8 Mar 2019

by Zebari et al. The responses are given in "*Italic*" font style.

Anonymous Referee #1

The authors of this manuscript try to use geomorphic indices and results of landscape modelling to constrain the relative timing of uplift of three anticlines. In general, the topic is interesting, and it will be a substantial contribution to the journal. Nevertheless, the revisions including the methodology and discussion, as well as the rearrangements of sections are still needed before publication. Major comments and suggestions are listed below.

1. Introduction: the authors should clearly state the importance of this study. Why the detailed spatial and temporal distribution of deformation ... is not yet well understood? Due to the lack of subsurface data, and/or this region is inaccessible for field surveys?

*Authors: It is not well understood due to the lack of comprehensive studies, insufficient surface and subsurface data, as well as access problems because of geopolitical conflicts. The clarification is given in new version of the manuscript (Section 1, Page2, Lines 6-7).*

2. Section 3.1.1: with aim of assessing landscape maturity along thrust-related anticlines, hypsometric curves and integrals have often been used for (sub-) drainage basins. The methodology differs from the three incomplete hypsometric curves displayed in Fig. 7. Actually, the authors did not extract drainage basins even if the stream channels of the Harir anticline have been shown in Fig. 6a.

*Authors: The hypsometric curves are now recalculated following the method of Pérez-Peña et al. (2009) as total weighted mean of hypsometric curves for all drainage basins that have an area of more than 0.25 $km^2$ within each anticline. This recalculation is given in the new version of the manuscript (Section 3.1.1, Page 5, Lines 7-10).*

3. Section 3.1.5 Digital elevation models: this section does not belong to the 3.1 geomorphic indices.

*Authors: Resolved by renumbering the sections (Section 3.2, Page 7, Lines 10-28).*

4. Section 5.1: the authors just described the rock erodibility. They should be included in geological setting, instead of discussion part. Here, the authors stated, "the stratigraphic column in the area consists of rocks with different erodibility" (page 11, line 29), and also mentioned in the conclusion "Due to the similarity in the lithology, structural setting and climate" (page 15, line 23-24). They should clearly state whether the difference exists or not.

*Authors: We removed this section. Information on rock erodibility is now included in the section describing the geological setting. In Section 5.1 (page 11, line 29) we mean that there is vertical variation in the rock erodibility in the stratigraphic columns. We have resistant Cretaceous and Paleogene interval of carbonate rocks and less resistant Upper Cretaceous-Tertiary intervals of clastic rocks. In the conclusion (page 15, line 23-24), we refer to the lateral extent of these stratigraphic units along the three anticlines, which is similar. We made the distinction between vertical variations in the rock erodibility and the lateral similarity in the exposed rock units clear in the new version of the manuscript to prevent confusion.*

---

## Author Comment (AC2) · 8 Mar 2019

by Zebari et al. The responses are given in "Italic" font style.

Anonymous Referee #2

Comments to the manuscript entitled "Relative Timing of Uplift along the Zagros Mountain Front Flexure Constrained by Geomorphic Indices and Landscape Modelling, Kurdistan Region of Iraq" by Zebari et al. (doi:10.5194/se-2018-124).

The authors of this manuscript try to constrain the relative timing of uplift of three anticlinal folds of the Iraqi Zagros Mts., combining the results of landscape evolution models and geomorphic indices. The topic fits the ones of the journal and the manuscript has the potential to be interesting for the international scientific community. Nonetheless, some general comments and minor specific ones are listed below, suggesting that some important revisions are needed before publication.

General comments:

1) Considering the deformation style of the Folded Zone of the Zagros Mts. chain, the assumption of constant rock uplift seems too simplistic. Doesn't the evidence of NW-ward propagation of the Harir Anticline (used by the authors for supporting the scenario of independent and diachronist uplift in different fold segments) affect the assumption of homogeneous and constant uplift rate in each structure? In this frame, the hypsometric analysis performed for the entire anticlinal ridges seems to have no sense, while I would suggest using an approach to the hypsometric analysis such as the one proposed by Pérez-Peña et al. (2009).

*Authors: We built our model to estimate the time it will take the present day Harir Anticline to reach the maturity level of the Akre Anticline under the assumption of constant climate and uplift rate. Then, we assume if the climate and uplift were the same in the past, the Akre Anticline has started to uplift before Harir Anticline in that estimated time. For this reason and for the sake of simplicity we used a constant uplift rate in the model.*

*We do not use absolute values of the geomorphic indices in our analysis, but we try to distinguish areas with relatively high values from those with relatively low values. Therefore, we think there is no need for applying the approach proposed by Pérez-Peña et al. (2009), because both the approach of Pérez-Peña et al. (2009) and our approach give similar results when it comes to defining areas with high and low (hypsometric integral, HI) values. Pérez-Peña et al. (2009) calculated HI for the given area with specific grid sizes (500m, 1km and 2km) and then calculated Moran's I index for each case to detect autocorrelation patterns in the data distributions. They conducted hot spot analyses by using the Getis-Ord Gi\* analysis within a specified distance (5km) to map the clusters of high and low HI values. In our approach, we calculated HI (and other indices) for each pixel by including the surrounding data of a specified distance based on the size of the used moving window. This means the results already have a relation with the surrounding cell without using Moran's I index and Getis-Ord Gi\* analysis. In the example given here (see the Figure below), first, we calculated HI from SRTM (30m resolution) by using a moving window of 3\*3 km and resample it to a 500m grid, then we used the approach by Pérez-Peña et al. (2009) for the same SRTM data with a grid of 500m and conducted Moran's I index and Getis-Ord Gi\* analysis for the cluster using 1.5 km distance. The results are similar in aspect of defining clusters of high and low HI values. We added this information to the manuscript.*

[Figure]

*a) HI calculated from the SRTM (30m) 100 * 100 cell (3*3 km) moving window and resampled to 500 m cells; b) Getis-Ord statistic estimation with 1.5 km distance for the HI that was calculated from the SRTM (30m) and with a 500 m grid following the method described by Pérez-Peña et al. (2009).*

2) Also the use of the only equations for fluvial erosion and diffusion processes for landscape evolution modelling may be too simplistic. Besides the justifications provided by the authors in section 5.2, it sounds not realistic that sedimentation on the slopes of anticlines can be neglected over the time-span of landscape evolution modelling (105 years), as well as the assumption of constant erosion rates (and climate!).

*Authors: The use of equations for fluvial erosion and diffusion processes only and neglecting the sedimentation in the model is explained in section 3.2. In such a landscape with steep slopes, the detachment-limited erosion due to the fluvial system tends to be the dominant process. Even if there is some fluvial sedimentation, it will be too small to affect the landscape. In addition, no notable sites of fluvial deposition were found on the anticline and its flanks from the field investigation and from satellite imagery. We add this information in the new version of the manuscript to clarify the reason for neglecting slope deposition (Section 3.3 Lines 5-9). In the model, the applied parameters were constant, while the erosion rate varies with the instant upstream drainage area and instant slope following the stream power incision law. The climate was kept constant for the sake of simplicity. The average of modeled paleo-precipitation anomalies for at least the last 300 kyr in the Eastern Mediterranean was close to zero as seen from the record of Lake Van in Turkey, some 200 km NNW of our study area (see the Figure below). Furthermore, considering the dimensions of the area, the changes in climate, if any, would have been constant over the entire study area, hence, affecting the three anticlines in the same way.*

[Figure]

*Precipitation anomalies for Lake Van, Turkey, for the last 300 ka (data were obtained from Mona Stockhecke by personal communication; Stockhecke et al., 2016).*

3) The authors justify the choice of the present topography of the Harir Anticline as LEM input asserting that in this structure the evolved drainage network overprinted the pre-existing one. Looking at Fig. 6 it seems that the drainage has a pattern similar to the one described by Ramsey et al. (2008; Basin Research (2008) 20, 23–48, doi: 10.1111/j.1365-2117.2007.00342.x) as evidence of lateral propagation of folds in the Zagros. This implies a diachronic fingerprint in the drainage network which could void the sense of performing the hypsometric analysis for the entire anticlinal ridges.

*Authors: In order to resolve this problem and in order to overcome the limitations of performing the hypsometric analysis for the entire anticlinal ridges, we recalculated the hypsometric curves as total weighted mean of hypsometric curves for all drainage basins that have an area of more than 0.25 km² within each anticline. The manuscript was modified accordingly (Section 3.1.1, Page 6, Lines 7-9; Figs. 6, 9. And 13).*

4) Some of the units stratigraphically above the Cretaceous limestones outcropping on the anticlines' crest are transitional to continental (i.e. the Bakhtiari Fm.), thus likely being affected by lateral variability of thickness. What about the effects on the uplift rate calculation based on thickness? Furthermore, this uplift rate was calculated based on the thickness and elevation of units on the anticline crest, but (again) is it correct to extend such a rate to the entire folds given their lateral growth?

*Authors: Since we have calculated the uplift rates based on the exhumed and exposed thickness since the onset of the MFF at 5 Ma, the variation in thickness of the units overlying the Cretaceous carbonates will affect the used uplift rate in the model. This thickness varies in between these three anticlines and even along strike within one anticline. The thickness of the units overlying the Cretaceous carbonates in the area ranges from ~ 2.0 km to maximum of 2.7 km. The uplift rates calculated based on these thicknesses will be in between 0.0007 to 0.0008 mm/yr. This range is not significant to noteworthy effect on the result of our model. We used the thicknesses found in well Bijeel-1, which is in a central part with respect to the three anticlines, to calculate the uplift rate. There will be variation also along a single anticline and we tried to overcome this by neglecting the two ends of anticlines in our analysis, and here our scope is to make a comparison in between the three anticlines omitting changes along a single anticline. We have clarified this in the new version of the manuscript (Section 4.3, Page 12, Lines 13-30).*

5) There are several repetitions over the manuscript (see "specific comments").

*Authors: We reviewed the manuscript and tried to remove the repetitions wherever they existed.*

6) Some original data (geological cross sections of Fig. 4) are referred to in the geological setting, while should be better described in the results.

*Authors: Fig. 4 is now moved to the result section and is described there (Section 4.3Page 12, Lines 13-30; Fig 10).*

7) In some cases, the interpretations seem not supported by data. For example, the fit between some of the hypsometric curves obtained with LEM and the ones computed for the three analyzed anticlines is not evident in Fig. 10 and the "minimum RMS" invoked by the authors to demonstrate the fit is not quantified. On the other hand, authors provide a quite specific timing for the inferred "delay" in the deformation sequence of the three folds which is based on this "fit" and use it to support the diachronic scenario of fold development. In my opinion such a constrain is weak, if based on the hypsometric analysis. Other doubt interpretations are listed in the specific comments.

*Authors: We don't find a better way of comparing the evolved landscape with that of the more mature anticlines in term of maturity rather than using hypsometric curves and other indices. Here, we used minimum RMS to find the closest curve statistically. Also, we have recalculated the hypsometric curves for output as the weighted mean of hypsometric curves for basins with area larger than 0.25 km2 within each anticlinal ridge weighted by the basin area within that anticline, and we have compared to that of the relatively more mature anticline. The specific time that we come calculate is the run time of the model that matches bests with the mature anticline. And with such model it is also difficult to assign a specific tolerance with ± and assign a margin of error. We provide the updated calculations in the new version of the manuscript (Fig 9).*

8) Section 5.1 doesn't sound necessary.

*Authors: We removed this section.*

9) In the Discussion new data are presented (i.e. Fig. 13), but it is not explained how they have been obtained, in particular the calculation of the slope/area. Is it obtained using just the drainage network or the whole topography?

*Authors: The data were obtained from the drainage network extracted from SRTM DEMs, so the slope is the stream slope at any specific point and the area is the upstream drainage area from that point. We added a short note in the methods about it in the new version (Section 3.2, Page 7, Lines 27-28).*

10) In the Conclusions authors refer to the three analyzed anticlines as "active folds", while in section 5.2 they state that the youngest unit affected by folding is the Mio-Pliocene Bakhtiari Fm.

*Authors: The Upper Bakhtiari Fm is also the youngest stratigraphic unit in the area. We point to this as the start for folding and consequent uplift and calculate the exhumed and exposed thickness of older strata in the area. When the Pliocene units are folded, it means that the folding should have been active in Pleistocene-Holocene (Section 5.1, Page 13, Lines 21-24).*

**Specific comments and technical corrections**

-TITLE: I would suggest to change the title into: "Relative Timing of Uplift along the Zagros Mountain Front Flexure (Kurdistan Region of Iraq): Constrains by Geomorphic Indices and Landscape Evolution Modelling"

*Authors: Done.*

- ABSTRACT:

PAGE 1, LINE 13: maybe "fold and thrust belt" and not "fault and thrust belt"

*Authors: Done.*

- INTRODUCTION:

PAGE 2, LINES 16-17: Why "The timing of this activity is expected to differ along-strike"? Any reference or explanation?

*Authors: In the literature, different times have been assigned to the onset of uplift in different part of the Zagros. This is explained in the next sentence in the manuscript by stating the activity period in the Iranian part of the MFF, referring to the corresponding citation (Section 1, Page 2, Lines 16-23).*

- GEOLOGICAL SETTING:

PAGE 4, LINE 31: change "river terraces" into "terraced alluvium".

*Authors: Here we mean aggregational river terraces that presents in varies elevated layers along the side of major rivers in the area. These sediments have been mapped and described under the term of "reviver terraces" (Jassim and Goff, 2006: Sissakian, 1997).*

PAGE 5, LINES 18-19: I suggest not to refer to new data in the geological setting. Fig.4 should be described (if made with newly surveyed data) in the Results.

*Authors: We have moved this to the result section (Section 4.3, Page 12, Line 17-30).*

- DATA AND METHODS:

PAGE 5, LINE 26: the hypsometric curve is not an "index"

*Authors: Separated.*

PAGE 5, LINE 27-29: the definition/meaning of the geomorphic tools is vague and in some cases, not correct (i.e. "The hypsometric curve and the hypsometric integral highlight raised and flat surfaces").

*Authors: Edited.*

PAGE 6, LINES 3-4: in general, the convex vs. concave shape of the hypsometric curve not necessarily reflects the "maturity" of a landscape (in terms of its absolute age) but can also depend on the type and rates of earth surface processes which dominate the landscape evolution (e.g. linear incision vs. hillslope diffusion processes).

*Authors: Rephrased.*

PAGE 6, LINES 10-11: again, the meaning of HI is not clearly defined. Please, rephrase.

*Authors: Redefined.*

PAGE 7, LINE 1: change the order of terms into "Nh, NHI and NSR"

*Authors: Done.*

PAGE 7, LINE 5: Digital Elevation Models (3.1.5.) are not Geomorphic Indices. This section should become 3.2

*Authors: Solved (Section 3.2, Page 7, Lines 10-28).*

PAGE 7, LINE 22: soil creep is mentioned as second main process inputed in LEM. Maybe the authors should refer more generally to hillslope diffusion processes.

*Authors: Done.*

PAGE 7, LINE 23-24: see general comment 2): it sounds strange that over the time-span of the modelling the sedimentation on slopes can be neglected.

*Authors: Here we mean sedimentation from the fluvial system. We clarified this meaning in the manuscript (see our response to the general comment 2 above).*

PAGE 8, LINE 8: again, the authors refer to "soil creep" (see comment above).

*Authors: Replaced.*

PAGE 8, LINES 17-18: see general comment 3).

*Authors: This may be the case for the majority of anticlines in the Zagros Belt; initiating from a segment, growing laterally, and then linking in the plunging end. This is addressed in the new version by recalculating the hypsometric curves as total weighted mean of hypsometric curves (see our answer to general comment 3).*

PAGE 8, LINE 30: authors refer to "time steps" before defining them.

*Authors: We redefined time step as follows: "In this approach, we ascribed values for m/n ranging from zero to one, and X was calculated for each time from Eq. 7".*

PAGE 9, LINE 13: "Kd = 0.001 m2yr-1": why exactly this value?

*Authors: Diffusivity coefficient varies with the thickness of soil (regolith) and since the soil is rare and very thin when it occurs, we assigned a low coefficient as explained by (Fernandes and Dietrich, 1997).*

- RESULTS:

PAGE 10, LINE 6: "HI values are maximum at the Greater Zab River": maybe authors mean that HI values are minimum?

*Authors: Typo; edited.*

PAGE 10, LINE 6-12: This part seems not necessary and the authors should pay attention to the meaning of HI when calculated for square areas and not for single basins. In this case HI measures how rapidly elevation changes and not strictly the amount of incision.

*Authors: We rephrased the paragraph.*

PAGE 10, LINE 13-18: results concerning roughness analysis are quite obvious...is it really necessary?

*Authors: We rephrased and reduced this section.*

PAGE 10, LINE 26-34: this part should be moved to the methodological section.

*Authors: It has been rephrased to match the results and unnecessary parts have been removed.*

PAGE 11, LINES 11-12: authors state that "In the landscape modelling, various simulations with different parameters and time spans were performed", nonetheless they do not provide details on the simulation (neither in the supplementary material). How did they select the best outputs?

*Authors: The results are added in the supplementary material and referred to in the manuscript.*

PAGE 11, LINES 13-14: authors state that "The evolving drainage system overprints the pre-existing one in the input and gradually becomes more deeply incised from the anticline flanks curving toward its core (Fig. 9)". This is not evident in Fig. 9, according to what already explained in the general comment 3).

*Authors: We think it is clear how the erosion is carving deeply toward the anticline core as it can be noticed from the contour map. Here, we show the 1000-contour line to show that (see the Figure below).*

[Figure]

*Black is the 1000-contour line from the input and red is the 1000-counter line from the output after 100 kyr.*

PAGE 11, LINES 15: change "plain" into "flat".

*Authors: Done.*

PAGE 11, LINES 20-22: see general comment 7)

*Authors: We addressed this by recalculating the hypsometric curves for output as the weighted mean of hypsometric curves for basins with area larger than 0.25 km² within each anticlinal ridge weighted by the basin area within that anticline, and we have calculated HI for the output and compared to that of the relatively more mature anticline (see our answer to the general comment 7).*

- DISCUSSION:

Is section 5.1 necessary?

*Authors: The section is removed.*

PAGE 12, LINES 14-15: what the authors mean with "The maturity level along these anticlines therefore represents the level when these carbonates cropped out in their latest stage"?

*Authors: The section is removed.*

PAGE 12, LINES 17-19: authors state that "A landscape survives when its uplift is not completely counterbalanced by erosion (Andreani and Gloaguen, 2016; Burbank and Anderson, 2012; Pérez-Peña et al., 2015)": it does not sound. . .maybe authors refer to relict landscapes?

*Authors: Yes, we mean that. We made this clear in the new version of manuscript to avoid confusion.*

PAGE 12, LINES 21-22: the sentence "The locations dissected by rivers show high surface roughness" seems obvious and not necessary.

*Authors: Removed.*

PAGE 12, LINES 29-31: "The same effect is visible in swath topographic profiles (Figs. 12c and 12d): in Harir Anticline, there is a clear topographic step with a higher slope angle, while in Akre Anticline the slope is gentler and more linear": to outline this evidence swath profiles are not necessary. . .if they can provide further evidence, the latter should be discussed.

*Authors: We removed these swath profiles.*

PAGES 12-13: "This can be interpreted with one of these premises: either both anticlines started to uplift successively (first Akre, then Perat, and finally Harir), or all of them started at the same time but with different uplift and exhumation rates (Akre the fastest, Harir the slowest)". This concept is repeated too many times over the manuscript. Furthermore, to justify the different geomorphic stage of the three folds with different uplift rates, shouldn't the latter be "fastest" in Harir and "slowest" in Akre??

*Authors: Due to high erodibility of the Upper Cretaceous – Lower Eocene succession in the area, it is flattened to the local base level by erosion wherever it crops out and does not make a notable topography. The indices measured for these anticlinal ridges, which are made of the Cretaceous carbonates, express the erosion action since the exposure of the carbonates. Based on these indices the Akre is relatively more mature than Harir, so the carbonate rocks must have been exposed to the erosion for a longer time than in Harir. Therefore, Akre's carbonates were at the surface before Harir's carbonate; either through an earlier onset of uplift or by a faster uplift rate, hence faster exposure to erosion.*

PAGE 13, LINE 13: How much does the assumption of constant rock uplift affect the results obtained? Since it is a "strong" assumption you should give an estimation of that.

*Authors: Since there are only data about the uplift rates in the area averaged over the last 5 Ma and for simplicity of the model, we had no option rather than applying a constant uplift rate. To know the effects of constant (linear) uplift on the result, we need to apply nonlinear uplift rates and compare the results. Using any nonlinear uplift rates will be arbitrary. The uplift rate used in our model was estimated from the exhumed and/or exposed thickness from the onset of the MFF.*

PAGE 13, LINE 15-26: This part of the discussion is not so clear. E.g. how did the authors perform the slope/area ˘ analysis?

*Authors: The slope/area curve was obtained as an average trendline for the streams in each anticline. The slope is for specific points at defined distance along the stream; where area is the upstream watershed area at each point. It is now better explained in the manuscript (Section 3.2, Page 7, Line 27-28; Section, 5.1 Pages 13-14, Line33-34 and 1-4).*

Some statements seem wrong: e.g. "In the Akre Anticline, this relationship [slope/area] is negative (Fig. 13b), which means that the streams have a concave shape and the segments with steeper slopes have migrated toward the core of the anticline. This implies that tectonic activity in the Harir Anticline is younger than in the Akre Anticline. Therefore, the premise of having Harir Anticline starting its uplift later than Akre Anticline is most likely". Why a higher uplift rate in the Harir couldn't have caused the same effect?

*Authors: We mean that the Harir Anticline was not exposed to erosion for a longer time, so the streams have not had as much time to carve deep in to the anticline as at Akre. Considering a unique onset, if Harir had a higher uplift rate, it would be exposed to erosion earlier than Akre, hence experienced more maturity, but it did not. Considering different onsets, even with a higher uplift rate, Harir should have been exposed to erosion later than Akre to have less maturity as it does.*

"Since the Upper Cretaceous carbonates in Harir Anticline were exposed later than in Akre Anticline, a landscape evolution model is a viable approach to estimate the exposure time difference. Here the model is built for the first premise of different onsets of uplift. Even if the second premise of different uplift rates is correct, the estimated time difference of the carbonate exposure will only be 28% less than that for the first scenario. As described in section 4.2, the calculated uplift time difference

between Akre and Harir anticline is 200±20 kyr, and if the second scenario is correct, the time difference of the carbonate exposure would be 144±14.4 kyr" This sentences are confused and the interpretation is not clear and a bit circular (choice of scenario based on modelling, based on constant uplift rates. . .).

*Authors: Since the first premise is better evidenced from the present-day morphology of both anticlines, we do not need to add an additional assumption based on the second scenario, so we have removed this part from the manuscript to avoid confusion.*

PAGE 13, LINE 27-34: see general comment 2).

*Authors: We addressed this in new version of the manuscript as explained in our answer to general comment 2 above.*

PAGE 14, LINE 8: The variations in stratigraphic thickness in between the anticlines is constrained by field data? And how does this variability affect the calculations of uplift rates?

*Authors: Since we have calculated the uplift rates based on the exhumed and exposed thickness since the onset of the MFF at 5 Ma, the variation in thickness of the units overlying the Cretaceous carbonates will affect the used uplift rate in the model. This thickness varies in between the three anticlines and even along strike within one anticline. The thickness of the units overlying the Cretaceous carbonates in the area ranges from ~ 2.0 km to maximum of 2.7 km. The uplift rates calculated based on these thicknesses will be in between 0.0007 to 0.0008 m/yr. This range is not noteworthy to effect on the result of our model. For this we used thickness in the well Bijeel-1, which is in a central part with respect to the three anticlines, to calculate the uplift rate. There will be variation also along a single anticline, we tried to overcome this by neglecting the two ends of anticlines in our analysis, and here our scope is to make a comparison in between the three anticlines omitting changes along a single anticline. We made this more clear in the new version of the manuscript (Section 4.3, Page 12, Lines 17-30).*

PAGE 14, LINE 13-26: Is this part really necessary and functional?

*Authors: This paragraph is added in order to explain the tectonic framework of further propagation of deformation in the SE side of the Greater Zab River and presence of another anticline to the south and southeast of Harir Anticline. And to explain the fold relay and to make sure that the uplift was not continuous from the NW to SW but rather in separated segments that propagated NW-ward until folds had linkage with each other.*

- CONCLUSIONS: See general comment 10

*Authors: The answer is provided under general comment 10 above.*

FIGURES:

FIG.1: i) I suggest to change "transform fault" into "strike-slip fault" in legend; ii) I suggest to add a legend for the different colors in transparency (e.g. the rose one corresponds to 2 zones. . .); check the use (or not) of parenthesis in the naming of zones;

*Authors: i) Done, ii) the colors in transparency characterize different morpho-tectonic zones of the Region. The names of these zones are given on the map, hence there is no need for a legend. The names within the parenthesis are equivalent names of these zones in the Iranian Zagros in the literature. This is made clear in the manuscript.*

FIG.2: i) I suggest to draw the border between different units in map; ii) check the numbering of the figures recalled in the sketch (they do not correspond to the figure recalled)

*Authors: i) Done, ii) Done.*

FIG.4: move and comment it in the Results.

Authors: Done.

FIG.5: symbols for wind gap and limit of Cretaceous limestone outcrop are not visible in figure Fig 6.

*Authors: We have updated and improved all figures in the manuscript.*

FIG.6: Symbols for the strike-slip component of displacement are missing

*Authors: There are no strike-slip faults within the Fig. 6.*

FIG.7: i) I suggest to prepare a new figure after having used the approach by Pérez-Peña et al (2009) to the hypsometric analysis.

*Authors: Same response as in the general comment 1; a new figure was added to the supplementary material, covering the entire study area.*

FIGs.8, 9, 10: to be revised according to the comments on the analytical procedures.

*Authors: These Figures were revised, accordingly.*

---

## Referee Report (RR1)

[revised manuscript text omitted]

The structure of fold-faults varies laterally and not homogeneously. The Perat anticline is a pop-up structure whereas the Harir and Akre anticlines are fault -propgation folds. In total, the eroded volumes are not the same: the lower Cretaceous dissected on the anticlinal hinge of Akre (strong erosion) differs from Harir anticline where the Cretaceous is little eroded.

Based on these geomorphic indices of the three anticlines we conclude that there is a measurable difference in landscape maturity between them. The difference in the maturity level must be due to a difference in one or more of the factors tectonics, climate, or rock erodibility. No variation in the climate is expected along the scale of these anticlines, therefore its impact on the landscape maturity can be neglected. The three anticlines show essentially the same lithology (Figs. 2 and 3). Thus, the only factors that may vary along the anticlines are uplift rate or onset of the uplift. This can be interpreted with one of the following scenarios: either the anticlines started to uplift in the order (1) Akre, (2) Perat and (3) Harir from west to east, or all of them started at the same time but with different rates. In the latter case, the uplift rate would have been highest at Akre and lowest at Harir.

**4.2 Landscape Model**

The aged landscape from the model run is the result of fluvial erosion and hillslope diffusion on the one hand, and uplift due to folding on the other hand. In the landscape modelling, various simulations with different parameters and time spans were performed. Harir Anticline was used as an input model and the landscape evolution model was run for a time span of 10 kyr up to 100 kyr and then it was run for a time span of 20 kyr. The evolving drainage system overprints the pre-existing one in the input and gradually becomes more deeply incised from the anticline flanks curving toward its core (Fig. 9). Harir Anticline is a box-shaped anticline with a wide and plain crest area. With ongoing incision towards the core of the anticline, this plain crest narrowed gradually and finally became a sharp ridge that divided the drainage basins on the SW flank from those in the NE.

We compared the hypsometric curves of the model outputs to the present-day curves of the anticlines (Fig. 10). Statistically, the hypsometric curve of Harir Anticline was closest to the present-day Perat Anticline after 70 kyr of erosion. The output curve after 200 kyr matched best with present-day Akre Anticline (with minimum RMS). We conclude that it will take Harir Anticline about 70 kyr to reach the maturity level of Perat Anticline and 200 kyr to reach the level of Akre Anticline if the uplift rates of the three anticlines were the same. The other possibility is that the anticlines started to grow at the same time but with different uplift rates. In this case, it is not possible to find the difference in uplift rates via our landscape modelling. Since the factors that control geomorphology (lithology, structural setting, and climate) were similar for all three anticlines, and under the assumption of constant growth and erosion rates, we infer that uplift of Akre an Perat anticlines started respectively 200±20 kyr and 70 kyr before Harir started to grow if their uplift rates were the same.

In the study of the recent deformation of the Folded zone of the Zagros Mts, the assumption of constant rock uplift and erosion seems too simplists.

[revised manuscript text omitted]

A systematic morphometric study (statistical computation) on all valleys using the same variables (elevation ...) was more relevant than the use of two insignificant examples. Why not use other parameters: longitudinal profiles of streams intersecting anticlinal flanks (knickpoints), Ksn, ...

[Figure]

[Figure]

[Figure]

**Figure 13: a) Diagram showing the exposure time of the Upper Cretaceous carbonates in Akre and Harir Anticlines. Two different scenarios are plotted for Harir: Having a slower uplift rate than Akre, or onset of uplift later than Akre. b) Channel slope-drainage area plots of streams in both Akre and Harir Anticlines.**

[Figure]

**Figure 14: Hypsometric curves for the studied anticlines as compared to those of the Shakrok and Safin anticlines, which show that the Harir's curve is more convex than that of both Shakrok and Safin.**

[Figure]

**Figure 15: Simplified history of the formation of anticlines during the propagation of the deformation front over time in the study area. The Harir anticline is likely the latest to have formed within the High Folded Zone in its SE end. It occupies the position of a relay structure during the linkage of two adjacent, but overlapping segments of the deformation front.**

---

## Author Response (AR2)

[revised manuscript text omitted]

by Zebari et al. The responses are given in "*Italic*" font style.

Anonymous Referee #1_v2

Submitted: 27 March 2019

The authors have fully considered my previous suggestions, and the revised manuscript has been greatly improved. Very few comments/suggestions are listed as follows.

1. from page 11, line 30 to page 12, line 7, and lines 23-30, these sentences seem like discussion.

*Authors: The mentioned pages and lines above are located within the result section. Here we tried to give information about the presented results and explain where the results of landscape modelling came from.*

2. page 32, three separate figures marked by a, b, c are encouraged, similar to the following figures.

*Authors: Appreciate it.*

3. page 33, two figures are displayed. BTW, how to identify wind and water gaps? DEM-based ridge lines of the anticlines may help to clarify.

*Authors: The first one is already removed but it remained there due to tracking changes. The water gaps are identified wherever an anticline is crossed by a stream; whereas, the wind gaps are identified wherever an anticline is crossed by a dry valley (abandoned river). The valley is above the base of the anticline and formed when the anticline uplift was higher than the erosional cut down of the river, therefore the river deviates around the tip of anticline and leave an abandoned valley crossing the anticline. We identified the wind gaps and water gaps on the DEMs and in the field.*

4. page 39, check the fig. 9 again. Find better places for annotating the (a) and (b).

*Authors: Done.*

5. page 44-45, which is figure 14 in revised version. No fig. 15 mentioned in the text.

*Authors: The one in page 45 is figure 14. The figure in the page 44 is already removed but it remained there due to tracking changes. This problem is resolved in the last revised version.*

Response to the Anonymous Referee #3's Interactive comment on **"Relative Timing of Uplift along the Zagros Mountain Front Flexure Constrained by Geomorphic Indices and Landscape Modelling, Kurdistan Region of Iraq"**

by Zebari et al. The responses are given in "*Italic*" font style.

Referee #3 Bernard Delcaillau

Submitted: 21 March 2019

We appreciate the reviewer for his thoughtful comment on the manuscript. We have considered these comments carefully and we respond to the comments with taking the following points into consideration:

1.  The reviewer has commented on the initial version of the manuscript (submitted on 28.11.2018) before the first round of revision, therefore, some of the comments are not relevant to the revised manuscript (submitted on 13.03.2019). Some other comments have already been taken into consideration based on the comments of the first two reviewers.
2.  Some of the comments either request for another way of identifying the landscape maturity, which yields similar results to those of our approach, or they are not applicable to our work due to limitation of approaches to overcome these shortages, or they are not necessary based on our observation. We provide clear explanation/reasoning for this type of comments in our responses below.
3.  We carefully considered the remaining relevant comments and made relevant changes to the manuscript.

General comments:

1) In my opinion, it would be useful the index Normalized channel steepness index ($K_{sn}$) which determines the relative gradient of channels. In this work does not appear the slope map also essential.

*Authors: We have tested many other indices including normalized channel steepness index ($K_{sn}$), which is presented in the figure below, stream power index, and stream length-gradient index. All of them show almost the same results in aspect of the relative maturity between the three anticlines. The indices show that Harir is less mature and Akre is more mature. Therefore, we rely on the indices that are already presented in the manuscript. However, a slope map is added to the Figure 4, Page 27.*

*The normalized channel steepness index ($K_{sn}$) is calculated as (Wobus et al., 2006):*

$$K_{sn} = \frac{S}{A^{-\theta}}$$

*Where S is the local channel slope [m/m], A is the upstream drainage area [m$^2$] and is typically taken as a proxy for discharge, and θ is the channel concavity.*

[Figure]

*Figure: Normalized steepness index (Ksn) for streams in the studied anticlines. It shows that Ksn is low in the crest and higher in the flank of the Harir Anticline, while it is lower in the flanks and increase toward the crest in both Akre and Perat anticlines. Ksn around main rivers that cross the anticlines was not calculated to exclude their effect in analysis.*

2) The concept of maturity of landscape is to be redefined more precisely according to poorly presented criteria in the manuscript.

*Authors: In this manuscript we tried to asses relative maturity between the studied anticlines and make a comparison between them using the geomorphic criteria that are widely used in many other similar studies. As we explained in the comment no. 1, we obtained the same results in aspect of relative maturity between the three anticlines from most of the indices that are used for defining the landscape maturity. The relative maturity of the anticlines is clearly explained in the revised version based on the comments by previous reviewers.*

3) The use of the only equations for fluvial erosion and diffusion processes for landscape evolution modelling may be too simplistic. Parameters concerning erosion processes are lacking in the modelling of landscape evolution.

*Authors: The reason behind using only equations for fluvial erosion and diffusion processes and uplift in the model is explained explicitly in Section 3.3, Page 8, Lines 5-9. Chen et al. (2014) showed that consideration of only these two components is sufficient for many landscapes, but cannot model fluvial sedimentation. From field observations and from satellite imagery, we infer that no significant fluvial sedimentation takes place on the slopes of the analyzed anticlines. On slopes of anticline flanks, the detachment-limited erosion due to the fluvial system tends to be the dominant process (Howard, 1994). To detect changes in the landscape due to fluvial erosion through time, we applied the commonly accepted idea that the rate of stream incision is directly proportional to the hydraulic shear stress of a stream (Braun and Willett, 2013). Consequently, we used the stream power incision law (Sklar and Dietrich, 1998; Whipple and Tucker, 1999). To account for the provision of sediment due to hillslope diffusion processes from slopes outside the river system, we used the hillslope diffusion equation (Culling, 1963; Tucker and Bras, 1998). Finally, the uplift rate is accounted for as well, as it is subtracted by the changes due to both fluvial erosion and hillslope diffusion. Only these two components were used in most of the studies that relate the landscape evolution with*

*underlying tectonics (e. g. Braun and Sambridge, 1997; Collignon et al., 2016; Cowie et al., 2006; Koons, 1995; Maniatis et al., 2009; Miller et al., 2007; Refice et al., 2012; Yanites et al., 2017; and others). Therefore, we think that applying only these components does not affect the validity of our landscape evolution model. Besides, it makes our study easily comparable with similar studies elsewhere. Since the three anticlines are made up of the same lithology (carbonates), there is no necessity to account for varying lithology and erodibility in our model.*

*Regarding the parameter used in the model, some of them are obtained directly from the present-day topography of the input, e.g. power-law coefficients m/n. Others were obtained from the literature on areas that are comparable with our study area in the aspects of lithology and precipitation, because there are no in situ denudation data that are necessary for obtaining parameters like the erodibility coefficient K for our model. Actually, if the in situ comprehensive erosional data were available for streams in the flank of these anticlines, there was no need for landscape evolution modeling. We could use them directly to measure the time difference between the maturity (erosion state) of the studied anticlines.*

4) The relationship between the hydrographic network and the growth of active anticlines is insufficient. It lacks a more serious morphometric study (for example: geometry of the profiles along the rivers, measurement of recent denudation rates at the outlet of catchment draining basins by the cosmogenic 10Be detrital sediment method).

*Authors: Here we do not study the tectonic activity along specific rivers. That is why we used those indices that can quantify the whole study area. We also do not have denudation data; therefore, we used remote sensing data. However, we are well aware that the knowledge of denudation rates along rivers in that area would greatly improve our understanding of the tectonic activity (uplift) of the anticlines there, as correctly suggested by the reviewer. The measurement of denudation rates from cosmogenic [10]Be is beyond the scope of this study.*

5) Lack of bibliographic references: in the paleoclimatological domain.

*Authors: The paleoclimatological data are added in the supplementary materials and are referred in latest version of the manuscript Section 3.3, Page 9, Lines 22-24.*

Response to the in-line comments on the manuscript:

Section 1, Page 2, Lines 22-23: Others references

*Authors: There are many references on the topic, but here we cited only few of them for keeping it short.*

Section 3.1, Page 5, Line 27: This degree…

*Authors: We used the terms "proxies" and "relative" here along with using the indices for assessing the landscape maturity. Thereby we already left possibility for some uncertainties.*

Section 3.1.3, Page 6, Lines 18: (e.g. Hobson, 1972)

*Authors: We added the reference.*

Section 3.2, Page 7, Line 18: !!!!

*Authors: rephrased.*

Section 3.2, Page 9, Line 11: very low uplift rate that you consider to be lineat in time !!!!

*Authors: There maybe misunderstanding about the used unit which is meter per year not millimeter per year. The uplift rate of 0.0007 m/yr or 0.7 mm/yr is a reasonable rate for the area and it matches with the vertical uplift in Kurdistan that has been presented by Tozer et al. (2019).*

Section 4.1, Page 11, Lines 1-8: The structure of fold-faults varies laterally and not homogeneously. The Perat anticline is a pop-up structure whereas the Harir and Akre anticlines are fault -propgation folds. In total, the eroded volumes are not the same: the lower Cretaceous dissected on the anticlinal hinge of Akre (strong erosion) differs from Harir anticline where the Cretaceous is little eroded.

*Authors: As we presented, the geometry of these anticlines varies laterally, but the shortening across, and vertical uplift along these anticlines does not vary significantly. The uplifted amount here is proportional to the shortening, which does not vary too much laterally. We calculated and added the shortening on Figure 10 to clarify this. The dissected amount of the Cretaceous carbonates is reflected by the landscape maturity. These carbonates are more dissected in the Akre anticline than in the other ones.*

*Regarding the eroded volume, both cross-sections across the Akre and Perat anticlines were constructed along Zinta and Bekhme gorges, respectively. Thus, the topographic profile cannot be used to estimate the eroded volume. Instead, we have plotted the topographic profiles across these two anticlines (dotted black line; Figure 10, Page 32) from nearby transects across the anticlines where topography is not affected by major rivers.*

Section 4.2, Page 11, Lines 23-26: In the study of the recent deformation of the Folded zone of the Zagros Mts, the assumption of constant rock uplift and erosion seems too simplists.

*Authors: The relevant paragraph has been modified in the revised manuscript. We have used this assumption for the sake of simplicity and due to the lack of available data.*

Section 5.2, Page 12, Line 20: What are your arguments? references?

*Authors: Here we mean spatial variation in climate between these anticlines, which are located within the same climate zone. In such a local scale (less than 100 km), the climate (precipitation) does not vary significantly.*

Section 5.2, Page 12, Line 23: What are your arguments?

*Authors: the argument behind this sentence is explained by Andreani et al., (2014) and Andreani and Gloaguen (2016) as cited.*

Section 5.2, Page 13, Lines 19-20: !!!

*Authors: Rephrased; we mean the exposure of the Cretaceous carbonates in the Harir Anticline was later than in the Akre Anticline.*

Section 5.2, Page 13, Lines 29-32: too simplistic

*Authors: The paragraph is already rephrased in the newer version of the manuscript. We refer to the climate data that show that there was not much variation in precipitation in the last 300 kyr ago, which is the upper limit of our model. Regarding the absence of consideration for both rock fall and karstification, these are the limitations of the most landscape evolution model studies today (as it is explained in general comment no. 3 and references therein, which lack consideration for rock fall and karstification). Also, we explained that karstification does not have a significant impact on the landscape based on the field observations and remote sensing data (lack of caves, dolines, karren, etc.).*

Section 5.3, Page 14, Lines 21-26: unclear

*Authors: Rephrased.*

Figure 4, Page 24: Indicate the shortening rates of the three folds

*Authors: The shortening of each anticline is calculated and added to the corresponding cross sections in Figure 10.*

Figure 12, Page 32: A systematic morphometric study (statistical computation) on all valleys using the same variables (elevation ...) was more relevant than the use of two insignificant examples. Why not use other parameters: longitudinal profiles of streams intersecting anticlinal flanks (knickpoints), Ksn, ...

*Authors: As we explained in the main comments no. 1 and 4, here we assess the relative landscape maturity between these three anticlines in general. Based upon a variety of methods we tested, the results were very close; therefore, we think there is no need for presenting more analysis (e.g. Ksn, Topographic profiles and/or knickpoints). In the last version we even removed unnecessary swath topographic profiles based on the recommendation of a previous reviewer.*

Figure 12, Page 34: blocking, Out-of-sequence.

*Authors: We are mainly focusing on the tectonic geomorphology based on surface data and landscape maturity. We are afraid that including bold structural and tectonic terms in our map will require additional detailed structural reconstructions that should be justified by data at depth such as detachment levels through seismic profile interpretations, which are beyond the scope of our manuscript.*